# Investigating the composition and recruitment of the mycobacterial ImuA'–ImuB–DnaE2 mutasome

Sophia Gessner[1,2†], Zela Alexandria-Mae Martin[1,2,3†], Michael A Reiche[1,2,4†], Joana A Santos[5‡], Ryan Dinkele[1,2], Atondaho Ramudzuli[1,2], Neeraj Dhar[3§], Timothy J de Wet[1,2,6], Saber Anoosheh[1,2#], Dirk M Lang[7], Jesse Aaron[4], Teng-Leong Chew[4], Jennifer Herrmann[8,9], Rolf Müller[8,9], John D McKinney[3], Roger Woodgate[10], Valerie Mizrahi[1,2,11], Česlovas Venclovas[12], Meindert H Lamers[5], Digby F Warner[1,2,11]*

[1]SAMRC/NHLS/UCT Molecular Mycobacteriology Research Unit, DSI/NRF Centre of Excellence for Biomedical TB Research, Department of Pathology, University of Cape Town, Cape Town, South Africa; [2]Institute of Infectious Disease and Molecular Medicine, University of Cape Town, Cape Town, South Africa; [3]Laboratory of Microbiology and Microsystems, School of Life Sciences, Swiss Federal Institute of Technology in Lausanne (EPFL), Lausanne, Switzerland; [4]Advanced Imaging Center, Howard Hughes Medical Institute, Ashburn, United States; [5]Department of Cell and Chemical Biology, Leiden University Medical Center, Leiden, Netherlands; [6]Department of Integrative Biomedical Sciences, University of Cape Town, Cape Town, South Africa; [7]Confocal and Light Microscope Imaging Facility, Department of Human Biology, University of Cape Town, Cape Town, South Africa; [8]Helmholtz Centre for Infection Research, Helmholtz Institute for Pharmaceutical Research Saarland, Saarbrücken, Germany; [9]German Centre for Infection Research (DZIF), Partner Site Hannover-Braunschweig, Braunschweig, Germany; [10]Laboratory of Genomic Integrity, Eunice Kennedy Shriver National Institute of Child Health and Human Development, Bethesda, United States; [11]Wellcome Centre for Infectious Diseases Research in Africa, University of Cape Town, Cape Town, South Africa; [12]Institute of Biotechnology, Vilnius University, Vilnius, Lithuania

*For correspondence:
digby.warner@uct.ac.za

†These authors contributed equally to this work

Present address: ‡Instituto de Investigação e Inovação em Saúde, Universidade do Porto, Porto, Portugal; §Vaccine and Infectious Disease Organization, University of Saskatchewan, Saskatoon, Canada; #Department of Chemistry and Umeå Centre for Microbial Research, Umeå University, Umeå, Sweden

Competing interest: The authors declare that no competing interests exist.

**Abstract** A DNA damage-inducible mutagenic gene cassette has been implicated in the emergence of drug resistance in *Mycobacterium tuberculosis* during anti-tuberculosis (TB) chemotherapy. However, the molecular composition and operation of the encoded 'mycobacterial mutasome' – minimally comprising DnaE2 polymerase and ImuA' and ImuB accessory proteins – remain elusive. Following exposure of mycobacteria to DNA damaging agents, we observe that DnaE2 and ImuB co-localize with the DNA polymerase III β subunit (β clamp) in distinct intracellular foci. Notably, genetic inactivation of the mutasome in an *imuB*[AAAAGG] mutant containing a disrupted β clamp-binding motif abolishes ImuB–β clamp focus formation, a phenotype recapitulated pharmacologically by treating bacilli with griselimycin and in biochemical assays in which this β clamp-binding antibiotic collapses pre-formed ImuB–β clamp complexes. These observations establish the essentiality of the ImuB–β clamp interaction for mutagenic DNA repair in mycobacteria, identifying the mutasome as target for adjunctive therapeutics designed to protect anti-TB drugs against emerging resistance.

## Editor's evaluation

This important study investigates the localization dynamics of the mycobacterial mutasome complex, comprised of ImuA', ImuB, and DnaE2. The mutasome complex has a key role in promoting mutagenic DNA replication during stress to increase the mutation rate and potential for selection of drug resistant mutations. The authors provide compelling evidence that ImuB localizes with the β-clamp upon damage exposure and that the clamp binding motif in ImuB is essential for its localization. These studies lay the ground for future work in this area and will be intriguing to a broad audience interested in bacterial physiology.

## Introduction

*Mycobacterium tuberculosis*, the causative agent of tuberculosis (TB), consistently ranks among the leading infectious killers worldwide (*World Health Organization, 2021*). The heavy burden imposed by TB on global public health is exacerbated by the emergence and spread of drug-resistant (DR) *M. tuberculosis* strains, with estimates indicating that DR-TB now accounts for approximately one-third of all deaths owing to antimicrobial resistance (*Hasan et al., 2018*). In the absence of a wholly protective vaccine, a continually replenishing pipeline of novel chemotherapeutics is required (*Evans and Mizrahi, 2018*) which, given the realities of modern antibiotic development (*Nielsen et al., 2019*), appears unsustainable. Therefore, alternative approaches must be explored including the identification of effective multidrug combinations (*Cokol et al., 2017*), the elucidation of 'resistance-proof' compounds (*Kling et al., 2015*), and the identification of so-called 'anti-evolution' drugs that might limit the development of drug resistance (*Smith and Romesberg, 2007*; *Ragheb et al., 2019*; *Merrikh and Kohli, 2020*).

Whereas many bacterial pathogens accelerate their evolution by sampling the immediate environment – for example, via fratricide, natural competence, or conjugation (*von Wintersdorff et al., 2016*; *Veening and Blokesch, 2017*) – these mechanisms appear inaccessible to *M. tuberculosis*: the bacillus does not possess plasmids (*Gray and Derbyshire, 2018*) and there appears to be no role for horizontal gene transfer in the modern evolution of strains of the *M. tuberculosis* complex (*Galagan, 2014*; *Boritsch and Brosch, 2016*). Instead, genetic variation in *M. tuberculosis* results exclusively from chromosomal rearrangements and mutations, a feature reflecting its ecological isolation (an obligate pathogen, *M. tuberculosis* has no known host outside humans) and the natural bottlenecks that occur during transmission (*Gagneux, 2018*). A question which therefore arises is whether a specific molecular mechanism(s) drives *M. tuberculosis* mutagenesis – perhaps under stressful conditions – and, consequently, if the activity thereof might be inhibited pharmacologically.

Multiple studies have investigated mycobacterial DNA replication and repair function in TB infection models (for recent reviews, *Singh, 2017*; *Minias et al., 2018*; *Mittal et al., 2020*). From these, the C-family DNA polymerase, DnaE2, has emerged as major contributor to mutagenesis under antibiotic treatment (*Boshoff et al., 2003*). A non-essential homolog of *E. coli* DNA Polymerase (Pol) IIIα (*Timinskas et al., 2014*), DnaE2 does not operate alone: the so-called 'accessory factors', *imuA'* and *imuB*, are critical for DnaE2-dependent mutagenesis (*Warner et al., 2010*). Both proteins are of unknown function, however *imuA'* and *imuB* are upregulated together with *dnaE2* following exposure of mycobacteria to DNA damaging agents including mitomycin C (MMC). That observation prompted the proposal that the three proteins might represent a 'mycobacterial mutasome' – named according to its functional analogy with the *E. coli* DNA Pol V mutasome comprising UmuD'$_2$C-RecA-ATP (*Jiang et al., 2009*; *Erdem et al., 2014*).

Here, we apply live-cell fluorescence and time-lapse microscopy in characterizing a panel of mycobacterial reporter strains expressing fluorescent translational fusions of each of the known mutasome components. The results of these analyses, together with complementary in vitro biochemical assays utilizing purified mycobacterial proteins, support the inference that ImuB serves as a hub protein, interacting with the *dnaN*-encoded mycobacterial β clamp and ImuA'. They also reinforce the essentiality of the ImuB–β clamp protein–protein interaction for mutasome function. Notably, while a strong ImuA'–ImuB interaction is detected in vitro, our live-cell data indicate the dispensability of either ImuA' or DnaE2 for ImuB localization – but not mutasome function – in bacilli exposed to genotoxic stress. Finally, using the β clamp-binding antibiotic, griselimycin (GRS) (*Kling et al., 2015*), we demonstrate in biochemical assays and in live mycobacteria the capacity to inhibit mutasome function through the

pharmacological disruption of ImuB–β focus formation. These observations suggest that, through its inhibition of β clamp binding, GRS might naturally limit the capacity for induced mutagenesis. As well as revealing a built-in mechanism protecting against auto-induced mutations to GRS resistance, our results therefore imply the potential utility of 'anti-evolution' antibiotics for TB.

## Results

### ImuB forms distinct subcellular foci under DNA damaging conditions

Our previous genetic evidence (*Warner et al., 2010*) informed a tentative model in which the presumed catalytically inactive Y family Pol homolog, ImuB, functioned as an adapter protein. According to the model, DnaE2 gains access to the repair site by interacting with ImuB, which similarly interacts with ImuA' and the *dnaN*-encoded β clamp subunit. To investigate the subcellular localizations of each of the mutasome proteins in live bacilli, we constructed reporter alleles in which the *M. smegmatis* mutasome proteins were labeled by N-terminal translational attachment of either Enhanced Green (EGFP) or Venus Fluorescent Protein (VFP) tags. The reporter alleles were introduced into each of three individual *M. smegmatis* mutasome gene deletion mutants – Δ*dnaE2*, Δ*imuA'*, and Δ*imuB* (*Warner et al., 2010*) – to yield the fluorescently tagged complemented strains, Δ*dnaE2 attB::egfp-dnaE2* (strain designated G-DnaE2, carrying *G-dnaE2* allele), Δ*imuB attB::egfp-imuB* (G-ImuB), and Δ*imuA' attB::vfp-imuA'* (V-ImuA') (*Figure 1—figure supplement 1A*).

The mycobacterial DNA damage response was induced by exposing the strains to the natural product antibiotic, MMC, an alkylating agent that causes monofunctional DNA adducts and inter- and intra-strand cross-links (*Bargonetti et al., 2010*). Following exposure of G-ImuB to MMC for 4 hr, distinct EGFP-ImuB foci were observed (*Figure 1A*). In contrast, a yellow fluorescence signal was observable throughout V-ImuA' cells, suggesting diffuse distribution of the VFP-ImuA' protein in the mycobacterial cytoplasm (*Figure 1B*). Although less distinct than G-ImuB, EGFP-DnaE2 produced similar evidence of focus formation in G-DnaE2 cells (*Figure 1C*). Notably, the significant increase in signal detectable in V-ImuA', G-ImuB, and G-DnaE2 cells following MMC exposure (*Figure 1—figure supplement 1B*) confirmed that expression of the respective fluorescence reporter alleles was DNA damage dependent in all three complemented mutants.

To ascertain if these observations were true for other types of DNA damage, the three reporter mutants were subjected to ultra-violet (UV) light exposure. Equivalent fluorescence phenotypes were observed for each of the three reporter alleles under both DNA damaging treatments (*Figure 1*). As UV exposure causes cyclobutane pyrimidine dimers or pyrimidine–pyrimidone (6–4) photoproducts (*Boshoff et al., 2003*), while MMC generates inter-strand DNA cross-links at CpG sites (*Tomasz, 1995*), these results indicated that expression and localization (recruitment) of the mutasome components might be independent of the nature of the genotoxic stress applied.

### N-terminal fluorescent reporters retain wild-type mutagenic function but are deficient in DNA damage tolerance

The addition of bulky fluorescent tags can disrupt the function of DNA replication and repair proteins (*Renzette et al., 2005*). To determine if any of the tagged mutasome proteins was affected, the functionalities of the *egfp-imuB*, *vfp-imuA'*, and *egfp-dnaE2* alleles were assessed in two standard assays (*Boshoff et al., 2003*; *Warner et al., 2010*): the first investigated DNA damage-induced mutagenesis by measuring the frequency of rifampicin (RIF) resistance following exposure to genotoxic stress, and the second tested DNA damage tolerance by spotting serial dilutions of each strain on media containing a DNA damaging agent. As observed previously (*Boshoff et al., 2003*; *Warner et al., 2010*), exposure of the wild-type parental *M. smegmatis* mc²155 to a sub-lethal dose of UV irradiation increased the frequency of RIF resistance 50- to 100-fold, as determined from enumeration of colony-forming units (CFU) on RIF-containing solid growth medium. In contrast, induced mutagenesis was greatly reduced in the Δ*imuA'*, Δ*imuB*, and Δ*dnaE2* deletion mutants, with mutation frequencies for these 'mutasome-deficient' strains approximately 20-fold lower than wild-type (*Figure 2A*). Notably, complementation with the cognate fluorescent reporter allele in V-ImuA', G-ImuB, and G-DnaE2 restored the UV-induced mutation frequencies of the three respective knockout mutants to near wild-type levels, establishing that each of the fluorescence reporter alleles retained function in UV-induced mutagenesis assays. In assays utilizing MMC instead of UV, a similar 20-fold reduction in

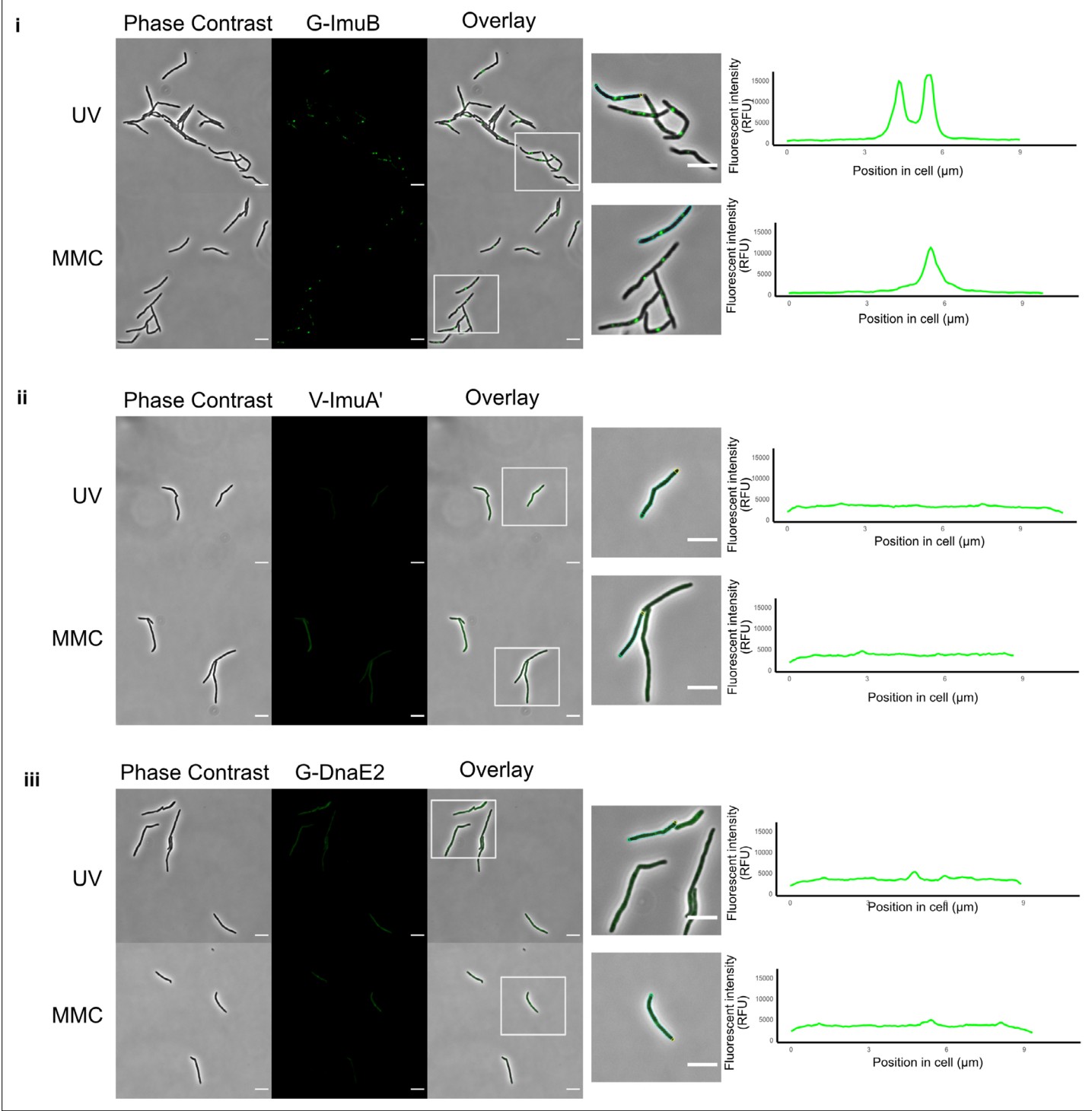

**Figure 1.** Visualization of the mycobacterial mutasome components. Representative stills from fluorescence microscopy experiments of *M. smegmatis* expressing translational reporters of the different mutasome components in their respective knockout backgrounds. Phase-contrast and fluorescence images of *M. smegmatis* expressing (**i**) G-imuB, (**ii**) V-imuA', and (**iii**) G-dnaE2 alleles are represented following 4 hr exposure to ultra-violet (UV) and 1× minimun inhibitory concentration (MIC) mitomycin C (MMC). White boxes indicate zoomed-in regions shown in the panels at right. The far right-hand panels indicate the fluorescence intensity determined along the longitudinal axis of a representative cell from each reporter mutant; the specific cell analyzed is outlined in the corresponding image to the left of the graph. Fluorescence microscopy experiments were repeated two to four times. Scale bars, 5 μm. Source data are available in Figure1.zip which can be accessed at http://doi.org/10.5061/dryad.76hdr7szc.

The online version of this article includes the following figure supplement(s) for figure 1:

**Figure supplement 1.** Design and representative fluorescent images of the mutasome translational reporters.

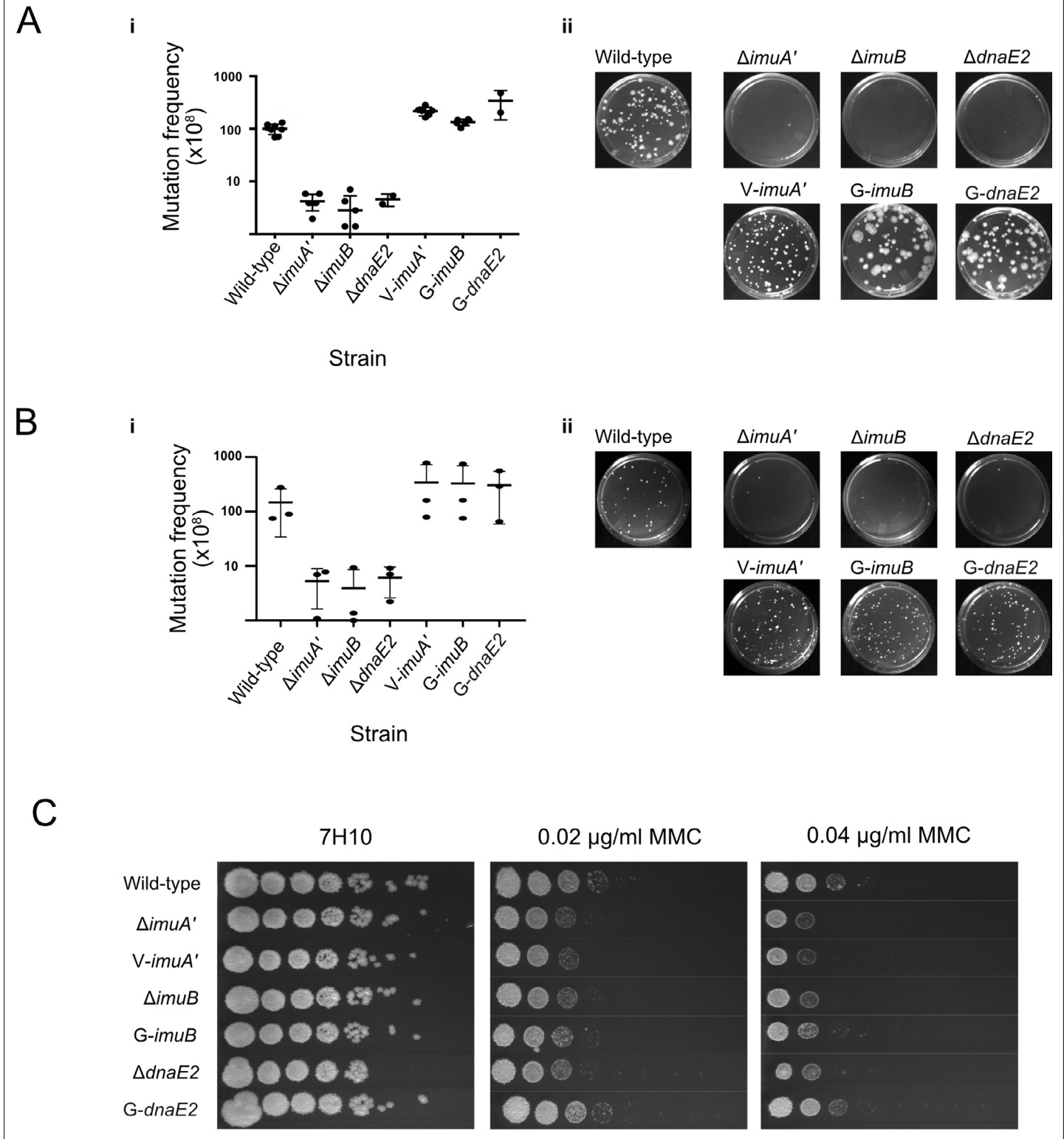

**Figure 2.** Functional validation of translational reporters. (**A**) N-terminally tagged fluorescence reporter mutants of M. smegmatis ImuA', ImuB, and DnaE2 retain function in DNA damage-induced mutagenesis. Cultures of *M. smegmatis* deletion mutants and complemented derivatives were exposed to 25 mJ/cm² of 254 nm ultra-violet (UV) light and allowed to recover for 3 hr before selection of rifampicin (RIF)-resistant mutants on RIF-containing 7H10 solid agar plates. (**i**) Mutation frequencies were calculated as a fraction of the CFU/ml of each culture prior to exposure to UV irradiation. Complementation with the corresponding fluorescence reporter alleles restored the resistance frequencies of the three mutasome knockout mutants (ΔimuA', ΔimuB, and ΔdnaE2) to levels observed in wild-type *M. smegmatis*. (ii) Representative RIF-containing plates with RIF-resistant mutants. (**B**) The

*Figure 2 continued on next page*

*Figure 2 continued*

same strains were exposed to 0.5× MIC mitomycin C (MMC) for 6 hr before plating on RIF-containing 7H10 solid plates. (**i**) Mutation frequencies were calculated as a fraction of the CFU/ml of each culture prior to exposure to MMC. As for the UV-induced mutagenesis assay, the fluorescence reporter alleles restored mutation frequencies to wild-type levels. (**ii**) Representative images of the RIF-containing plates with RIF-resistant mutants. (**C**) Serial dilutions of *M. smegmatis* deletion mutants and complemented strains were spotted on standard 7H10 and MMC-containing 7H10 plates. Results represent a minimum of three replicates for each strain. Source data are available in Figure2.zip which can be accessed at http://doi.org/10.5061/dryad.76hdr7szc.

The online version of this article includes the following figure supplement(s) for figure 2:

**Figure supplement 1.** V-ImuA' and G-ImuB fluorescent bioreporters restore ultra-violet (UV)-induced DNA damage sensitivities of the corresponding ΔimuA' and ΔimuB deletion mutants.

MMC-induced mutagenesis was observed in each of the three single knockout strains compared to wild-type, and this defect was restored when complemented with the respective fluorescent reporters (*Figure 2B*). In combination, these results confirmed the preservation of wild-type mutagenic function in the fluorescently tagged fusion proteins, irrespective of DNA damaging agent applied.

Surprisingly, the DNA damage tolerance assay – in which CFU-forming ability was tested during continuous exposure to MMC in solid growth media – produced contrasting results (*Figure 2C*): whereas the damage hypersusceptibility of the *dnaE2* knockout was reversed in the G-DnaE2 strain, complementation of either Δ*imuA'* or Δ*imuB* with its corresponding fluorescent reporter allele failed to restore a wild-type phenotype. The reason for these discrepant observations – restoration of both UV- and MMC-induced mutagenesis but not MMC-induced DNA damage tolerance – in the V-ImuA' and G-imuB strains is not clear. Although mutasome components are expressed in response to genotoxic stress arising from a variety of different sources, it is possible the different types and/or extent of DNA damage induced in the two separate assays used here (induced mutagenesis vs. DNA damage tolerance) might require distinct interactions with a different partner protein(s) and, further, that one/more of these might have been disrupted by the presence of the fluorescent tag(s). It is also plausible that, in the DNA damage survival assay, extended incubation in the presence of MMC (a clastogen with multiple effects on DNA integrity) might exacerbate the suboptimal operation of the mutasome owing to the presence of the bulky fluorophore – which differs significantly from the very brief exposure to the genotoxins in the induced mutagenesis assays. Consistent with the proposed impact of treatment duration on the functionality of the fluorescently tagged mutasome fusions, both V-ImuA' and G-ImuB mutants phenocopied wild-type in a UV damage sensitivity assay (*Figure 2—figure supplement 1*); however, these explanations are speculative and require further investigation. Given the inferred functionality of the fluorescence-tagged alleles in DNA damage-induced mutagenesis, we deemed them useful to investigate mutasome recruitment in live mycobacterial cells.

## ImuB localizes with the *dnaN*-encoded β clamp following DNA damage

We previously inferred that a putative interaction between ImuB and the *dnaN*-encoded β clamp was essential for mutasome function (*Warner et al., 2010*). To investigate the predicted interaction of ImuB and the β clamp in live bacilli, each of the three mutasome reporter alleles was introduced separately into an *M. smegmatis* mutant encoding an mCherry-tagged β clamp, mCherry-DnaN (*Santi et al., 2013*). The mCherry-DnaN reporter was chosen as background strain owing to its previous validation in single-cell, time-lapse fluorescence microscopy analyses of *M. smegmatis* replisome location (*Santi et al., 2013*; *Santi and McKinney, 2015*). For the time-lapse experiments, the resulting *M. smegmatis* dual reporter strains were grown in standard 7H9/OADC medium for 12 hr, following which the cells were exposed to MMC for 4.5 hr before switching back to 7H9/OADC for post-treatment recovery (*Figure 3*; *Videos 1–3*). At 4 hr post MMC treatment, distinct EGFP-ImuB foci were observed which, when overlaid with the mCherry-DnaN fluorescence signal, showed considerable overlap, suggesting association of the β clamp with ImuB (*Figure 3A, D*; *Video 1*). In addition to G-ImuB, the number of mCherry-DnaN foci also increased upon DNA damage (*Figure 3*; *Figure 3—figure supplement 1A*). In MMC-treated cells, the EGFP-ImuB signal was mostly detected in very close proximity to mCherry-DnaN foci (>50% of cells contained mCherry-DnaN and G-ImuB located within 0.3 μm of each other); almost the same frequency of association of mCherry-DnaN and G-ImuB foci was observed in bacilli exposed to UV, though the proportion of cells containing mCherry-DnaN foci alone was greater (*Figure 3A, D*; *Figure 3—figure supplement 1B*). In combination, these results are consistent with

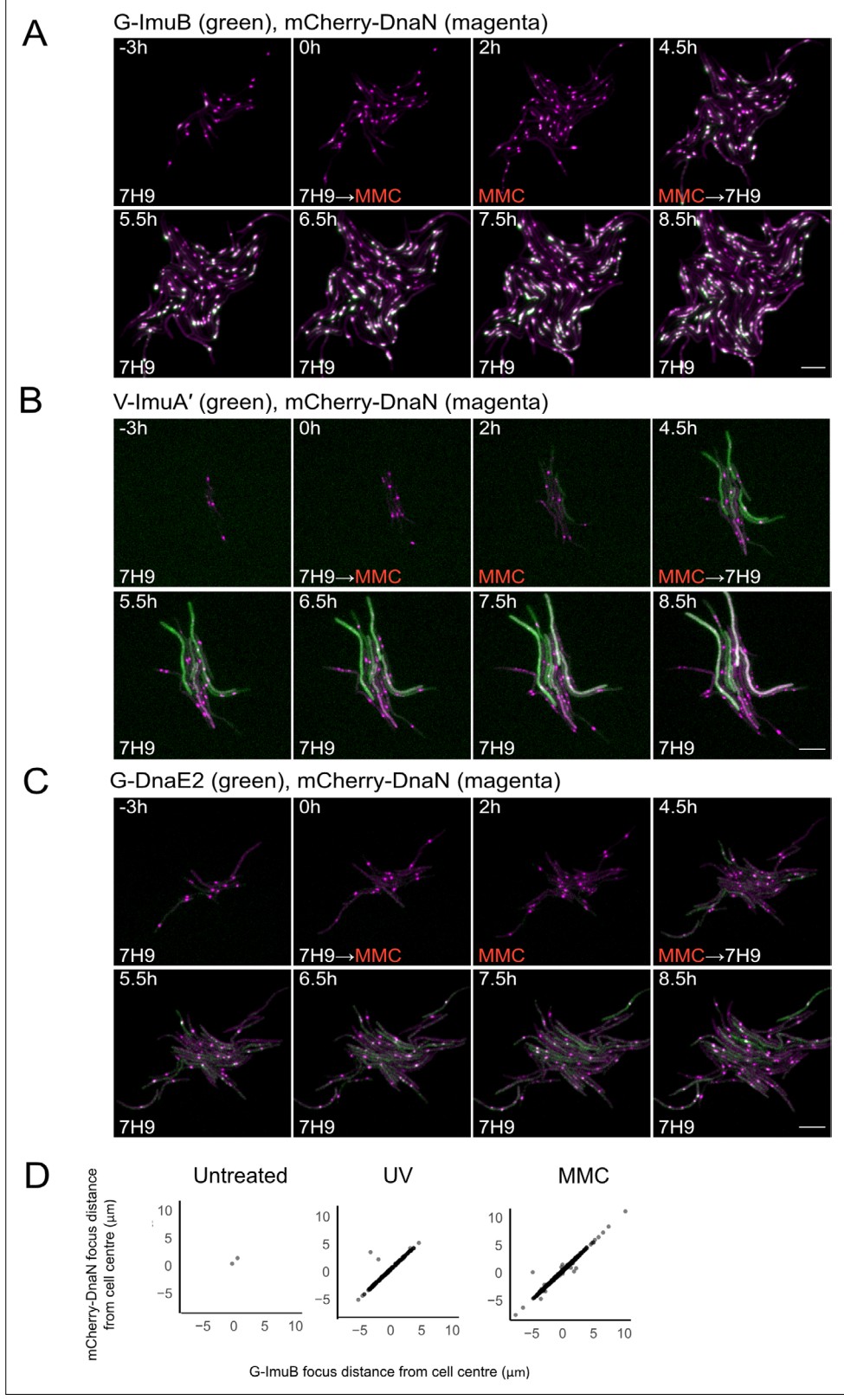

**Figure 3.** Representative time-lapse series of single cells of *M. smegmatis* expressing the mutasome reporters in combination with mCherry-DnaN. (**A**) G-ImuB (green) and mCherry-DnaN (magenta), (**B**) V-ImuA' (green) and mCherry-DnaN (magenta), and (**C**) G-DnaE2 (green) and mCherry-DnaN (magenta). Overlapping signals are viewed as white. The cells were exposed to 0.5× MIC MMC from time 0 hr until 4.5 hr, after which the medium

*Figure 3 continued on next page*

*Figure 3 continued*

was switched back to standard 7H9/OADC medium. Up to 80 XY points were imaged at 10-min intervals on fluorescence and phase channels for up to 36 hr. The experiments were repeated two to four times. Numbers indicate hours elapsed; scale bars, 5 µm. 7H9, Middlebrook 7H9 medium; MMC, mitomycin C. (**D**) Population-scale analysis of cells with both mCherry-DnaN foci and G-ImuB foci showed distinct overlap in location suggesting co-occurrence of the respective proteins. Source data are available in Figure3.zip which can be accessed at http://doi.org/10.5061/dryad.76hdr7szc.

The online version of this article includes the following figure supplement(s) for figure 3:

**Figure supplement 1.** mCherry-DnaN and G-ImuB focus formation following DNA damage treatment.

**Figure supplement 2.** Formation of fluorescent EGFP-ImuB foci in the absence of functional mutasome components.

---

the direct physical interaction of ImuB and the β clamp suggested previously by yeast two-hybrid and site-directed mutagenesis studies (*Warner et al., 2010*).

For V-ImuA′, a diffuse fluorescence signal was detected throughout the cells (*Figure 3B*; *Video 2*), rendering impossible any conclusion about the potential recruitment of ImuA′ to β clamp (mCherry-DnaN) foci. In contrast, the results for DnaE2 were more nuanced: overlap of peak fluorescence signals from EGFP-DnaE2 and mCherry-DnaN proteins was detected (*Figure 3C*) and was most evident within 1-hr post removal of MMC from the microfluidic chamber (*Video 3*). Although not as consistent as the ImuB–β clamp phenotype, the co-occurrence of DnaE2 and β clamp signals was reproducibly observed in multiple cells and across different experiments.

## ImuA′ and DnaE2 are not required for ImuB focus formation

We showed previously that deletion of *imuA′* phenocopied abrogation of either *imuB* or *dnaE2* in the MMC sensitivity assay (*Warner et al., 2010*) and, consistent with the interpretation that all three components are individually essential for mutasome activity, this phenotype was not exacerbated in a triple Δ*imuA′*–*imuB*–Δ*dnaE2* knockout strain. Together with yeast two-hybrid data which indicated a direct interaction between ImuB and ImuA′ (*Warner et al., 2010*), this observation raised the possibility that a deficiency in ImuA′ might impair ImuB protein localization. To test this prediction, the *egfp-imuB* allele was introduced into the Δ*imuA′* deletion mutant, generating a Δ*imuA′ attB::egfp-imuB* reporter strain. Despite the absence of ImuA′ in this mutant, EGFP-ImuB foci were observed following treatment with MMC (*Figure 3—figure supplement 2A*). Similarly, the absence of functional DnaE2 had no discernible impact on ImuB focus formation in either the site-directed *dnaE2*^AIA *attB::egfp-imuB* strain (*Figure 3—figure supplement 2B*) or the fully DnaE2-deleted Δ*dnaE2 attB::egfp-imuB* mutant (*Figure 3—figure supplement 2C*). In combination, these results appear to eliminate a role for either ImuA′ or DnaE2 in ImuB localization, instead implying the critical importance of the ImuB–β clamp interaction for mutasome assembly.

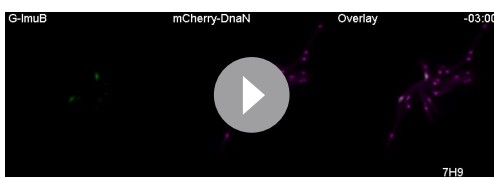

**Video 1.** Time-lapse microscopy of G-ImuB and mCherry-DnaN dual reporter. Representative time-lapse movie of the reporter strain expressing G-ImuB and mCherry-DnaN. Bacteria were imaged on fluorescence and phase channels for up to 36 hr at 10-min intervals. Treatment with MMC (100 ng/ml) was at 0–4.5 hr. This experiment was repeated six times. Numbers indicate the hours elapsed in the time-lapse experiment. 7H9, Middlebrook 7H9/OADC; MMC, mitomycin C. Scale bar, 5 µm. G-ImuB, green; mCherry-DnaN, magenta; overlay, white.

https://elifesciences.org/articles/75628/figures#video1

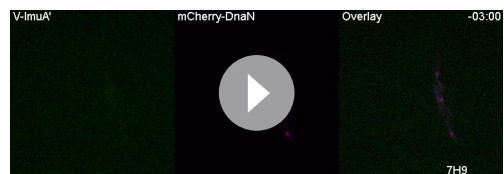

**Video 2.** Time-lapse microscopy of V-ImuA′ and mCherry-DnaN dual reporter. Representative time-lapse movie of the reporter strain expressing V-ImuA′ and mCherry-DnaN. Bacteria were imaged on fluorescence and phase channels for up to 36 hr at 10-min intervals. Treatment with MMC (100 ng/ml) was at 0–4.5 hr. This experiment was repeated three times. Numbers indicate the hours elapsed in the time-lapse experiment. 7H9, Middlebrook 7H9/OADC; MMC, mitomycin C. Scale bar, 5 µm. V-ImuA′, green; mCherry-DnaN, magenta; overlay, white.

https://elifesciences.org/articles/75628/figures#video2

**Video 3.** Time-lapse microscopy of G-DnaE2 and mCherry-DnaN dual reporter. Representative time-lapse movie of the reporter strain expressing G-DnaE2 and mCherry-DnaN. Bacteria were imaged on fluorescence and phase channels for up to 36 hr at 10-min intervals. Treatment with MMC (100 ng/ml) was at 0–4.5 hr. This experiment was repeated three times. Numbers indicate the hours elapsed in the time-lapse experiment. 7H9, Middlebrook 7H9/OADC; MMC, mitomycin C. Scale bar, 5 µm. G-DnaE2, green; mCherry-DnaN, magenta; overlay, white.
https://elifesciences.org/articles/75628/figures#video3

## Purified mutasome proteins interact in biochemical assays in vitro

All inference from this and previous work about the composition of the mycobacterial mutasome has been derived from microbiological assays. To address this limitation, we expressed and purified recombinant *M. smegmatis* mutasome proteins for biochemical analysis. Expression in *E. coli* of ImuB alone yielded low quantities of soluble protein that was prone to degradation, while attempts to express ImuA′ alone failed to generate soluble protein. In contrast, co-expression of ImuB with ImuA′ yielded both proteins in a soluble form (*Figure 4*). Subsequently, the ImuA′B complex could be captured via a histidine (His) affinity tag in ImuB. This confirmed that ImuA′ and ImuB interact in vitro, forming a stable complex even at protein concentrations as low as 400 nM (*Figure 4—figure supplement 1A*), corroborating previous yeast two-hybrid results (*Warner et al., 2010*). In *E. coli*, overexpression of DnaE2 resulted in insoluble protein, while DnaE2 overexpression in *M. smegmatis* appeared to be incompatible with cell viability: following transformation with the expression construct, very few colonies were obtained and could not be expanded in liquid culture (not shown).

Next, we analyzed the interaction of the *dnaN*-encoded β clamp with ImuB or the ImuA′B complex (*Figure 4*). Samples of the *M. smegmatis* β clamp with ImuA′B (*Figure 4A*, panel i) or ImuB (*Figure 4A*, panel ii) were injected into an analytical size-exclusion chromatography column and collected fractions subjected to sodium dodecyl sulfate–polyacrylamide gel electrophoresis (SDS–PAGE) analysis. Alone, the β clamp and ImuB/ImuA′B eluted at 1.47 and 1.54 ml, respectively. Incubation of the β clamp with either ImuB or ImuA′B caused a shift in the retention volume to 1.36 ml, indicative of complex formation. This was confirmed by SDS–PAGE analysis, which indicated co-elution of the β clamp with ImuB and ImuA′B (*Figure 4B*).

## EGFP-ImuB and VFP-ImuA′ form a stable complex

Our microbiological assays had unexpectedly revealed discrepant complementation phenotypes for the induced mutagenesis versus DNA damage tolerance assay (*Figure 2*), raising the possibility that the fluorescent tags in the bioreporter mutants might disrupt a protein–protein interaction(s) essential for DNA damage tolerance. We therefore investigated the capacity of the fluorescently labeled EGFP-ImuB and VFP-ImuA′ proteins to form a stable complex. To this end, His-EGFP-ImuB was co-expressed with Strep-VFP-ImuA′ in *E. coli* and the complex analyzed in three consecutive chromatography steps (*Figure 4—figure supplement 1B*). First, the cell lysate was loaded onto a HisTrap column to capture the VFP-ImuA′:EGFP-ImuB complex via the His-tag present in EGFP-ImuB. Next, the elution fractions containing the complex were loaded on a StrepTrap column to capture the complex via the strep-tag on VFP-ImuA′. Finally, the VFP-ImuA′:EGFP-ImuB complex was injected onto a size-exclusion column.

During all purification steps, EGFP-ImuB and VFP-ImuA′ were co-eluted as a complex, as indicated by SDS–PAGE analysis and fluorescent detection of EGFP-ImuB and VFP-ImuA′ in the same elution fractions. In combination, these observations suggest that the fluorescent tags did not disrupt ImuA′–ImuB complex formation in vitro – a result which implies that the absence in live cells of a clear ImuA′ (co-)localization phenotype was not attributable to the presence of N-terminal fluorophores.

## Inhibition of ImuB–β clamp-binding eliminates focus formation

Previous work established that the β clamp-binding domain of ImuB was essential for mutasome function: mutant strains carrying either a *imuB*$^{\Delta C168}$ allele (lacking the 168 amino acids in the ImuB C-terminal region) or a *imuB*$^{AAAAGG}$ allele (in which the wild-type β clamp-binding motif, $^{352}$QLPLWG$^{357}$, is substituted with the non-functional $^{352}$AAAAGG$^{357}$ peptide sequence) phenocopied full *imuB* deletion

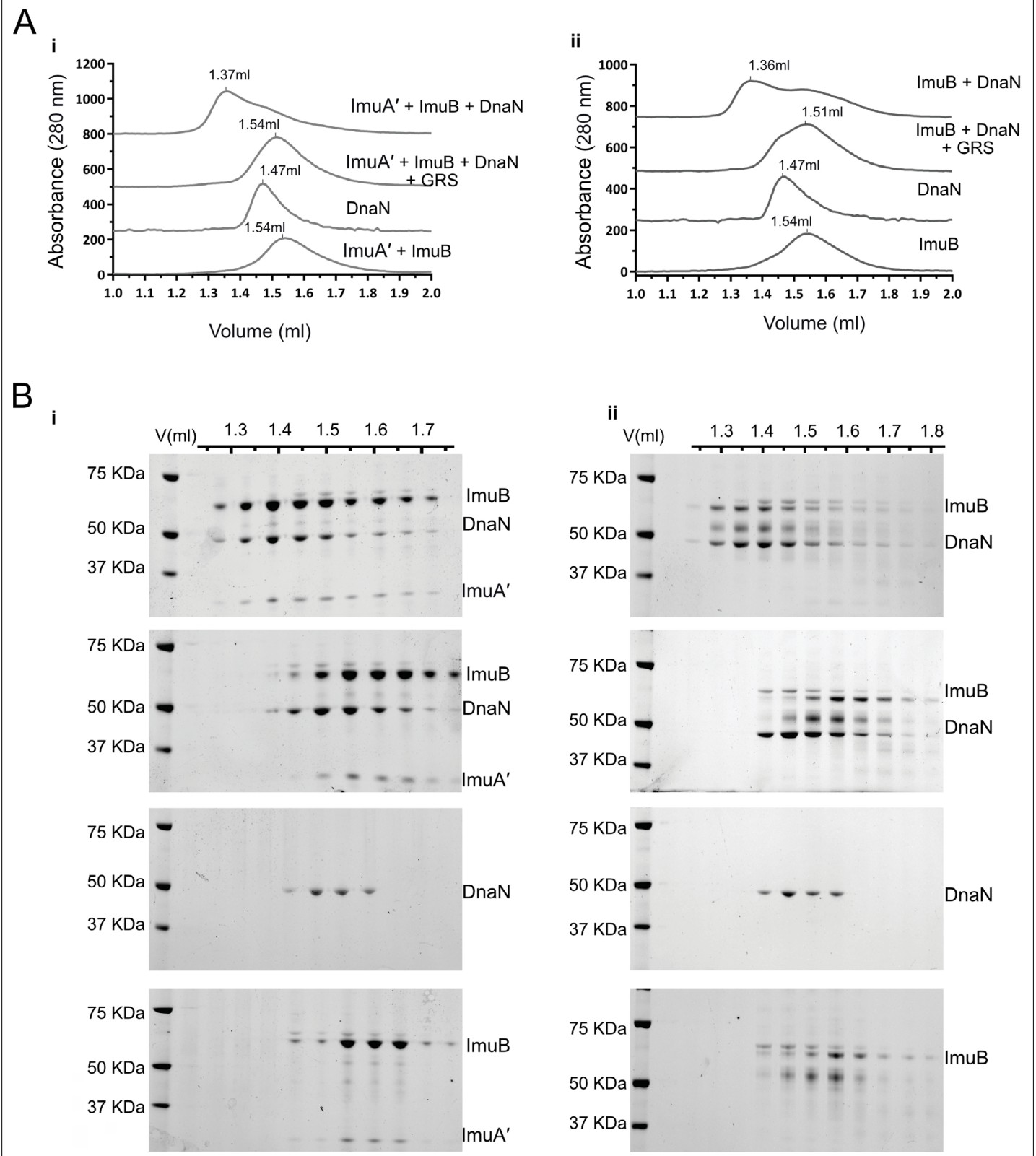

**Figure 4.** ImuB and ImuA'–ImuB interact with DnaN and these interactions are disrupted by griselimycin (GRS). (**A**) Gel filtration profiles of *M. smegmatis* (**i**) ImuA'B-DnaN and (**ii**) ImuB-DnaN complexes in the absence or presence of 15 µM GRS. For these experiments, 5 µM DnaN was added to 10 µM of (**i**) ImuA'B or (**ii**) ImuB. The gel filtration profiles of the individual proteins (ImuB and DnaN) or complex (ImuA'–ImuB) are shown for comparative purposes, and all curves were scaled for clarity. (**B**) Sodium dodecyl sulfate–polyacrylamide gel electrophoresis (SDS–PAGE) analysis of sequential

*Figure 4 continued on next page*

*Figure 4 continued*

fractions of the gel filtration runs. Gels are sorted in the same order as the corresponding gel filtration profiles shown in A. Source data are available in Figure4.zip which can be accessed at http://doi.org/10.5061/dryad.76hdr7szc.

The online version of this article includes the following figure supplement(s) for figure 4:

**Figure supplement 1.** Biochemical confirmation of stable ImuA'–ImuB complex formation and griselimycin (GRS)-binding DnaN.

---

(*Warner et al., 2010*). Therefore, to test the prediction that the recruitment of EGFP-ImuB and mCherry-DnaN into discernible foci was dependent on the ImuB–β clamp protein–protein interaction, we introduced an *egfp-imuB*^AAAAGG allele (G-*imuB*^AAAAGG) into the Δ*imuB* mutant. In contrast to the wild-type reporter (G-ImuB), the β clamp-binding motif mutant (G-ImuB^AAAAGG) exhibited no EGFP foci in any cell imaged following exposure to MMC (*Figure 5A*). Instead, the fluorescence was detectable throughout the cell as a diffuse signal. This result supports the inferred essentiality of the physical interaction between ImuB and β for ImuB localization and, moreover, establishes that detection of ImuB–β foci provides a reliable visual proxy for functional mutasome formation.

## GRS blocks ImuB–β clamp binding, preventing focus formation in *M. smegmatis*

GRS is a natural product antibiotic that binds the mycobacterial β clamp with high affinity, preventing DNA replication by blocking the essential interaction with the PolIIIα subunit, DnaE1 (*Kling et al., 2015*). Importantly, the region of GRS binding on β overlaps with the region predicted to interact with other β clamp-binding proteins (*Bunting et al., 2003*; *Burnouf et al., 2004*; *Kling et al., 2015*), including DnaE1 and ImuB (*Figure 5B*). From structural comparisons, this also holds true for *M. tuberculosis* (*Figure 5—figure supplement 1*). Therefore, we hypothesized that GRS might disrupt the ImuB–β interaction. Indeed, addition of GRS disrupted the in vitro interaction between the β clamp and pre-formed ImuA'B complex (*Figure 4A*, panel i) as well as between the β clamp and ImuB (*Figure 4A*, panel ii), as indicated by a gel filtration profile that is a superposition of the absorbance traces of the sample individual components (β clamp and ImuB or β clamp and ImuA'B). This was confirmed by SDS–PAGE analysis (*Figure 4B*). To confirm that the disrupting effect of GRS on the complex was the result of the GRS–β clamp binding (*Kling et al., 2015*), we measured the melting curves of the β clamp in the presence and absence of GRS (*Figure 4—figure supplement 1C*). Incubation with GRS led to a 3°C increase in the protein melting temperature, consistent with GRS binding to β. In contrast, GRS had no effect on the observed melting temperature of ImuA'B.

Finally, we examined whether these biochemical observations were recapitulated in vivo in live mycobacterial cells. To this end, the dual reporter mutant expressing mCherry-DnaN and G-ImuB was treated with GRS alone or following induction of DNA damage by UV or MMC exposure (*Figure 5*). Notably, the addition of GRS in combination with UV or MMC markedly reduced G-*imuB* focus formation (*Figure 5C*, panels ii, iii), with most cells phenocopying the diffuse fluorescence distribution observed following exposure of the β clamp-binding deficient EGFP-ImuB^AAAAGG mutant to UV or MMC (*Figure 5A*). Population analyses confirmed that GRS blocked ImuB focus formation in both UV- and MMC-exposed cells (*Figure 5D*), although the effect appeared more pronounced for the UV-damaged cells. The reasons for this difference are not clear. It is possible that the variety of mono-functional DNA adducts, inter- and intra-strand cross-links caused by MMC (*Bargonetti et al., 2010*) elicits a more profound DNA damage response than UV, which results in photoproducts and dimers; moreover, unlike UV, which is delivered as a transient exposure, MMC is a chemical clastogen which might persist inside mycobacterial cells before eventual elimination. Consistent with this proposal, it was evident that MMC treatment caused an elevated number of G-ImuB foci compared to UV – and GRS seemed more effective at reducing UV-induced mCherry-DnaN and G-ImuB foci than the corresponding MMC-induced foci, perhaps owing to 'trapping' of foci by MMC-induced DNA cross-links. Whatever the reason, the ability of GRS to inhibit G-ImuB focus formation in live mycobacterial cells exposed to two different DNA damaging agents suggested the potential for chemical disruption of mutasome function.

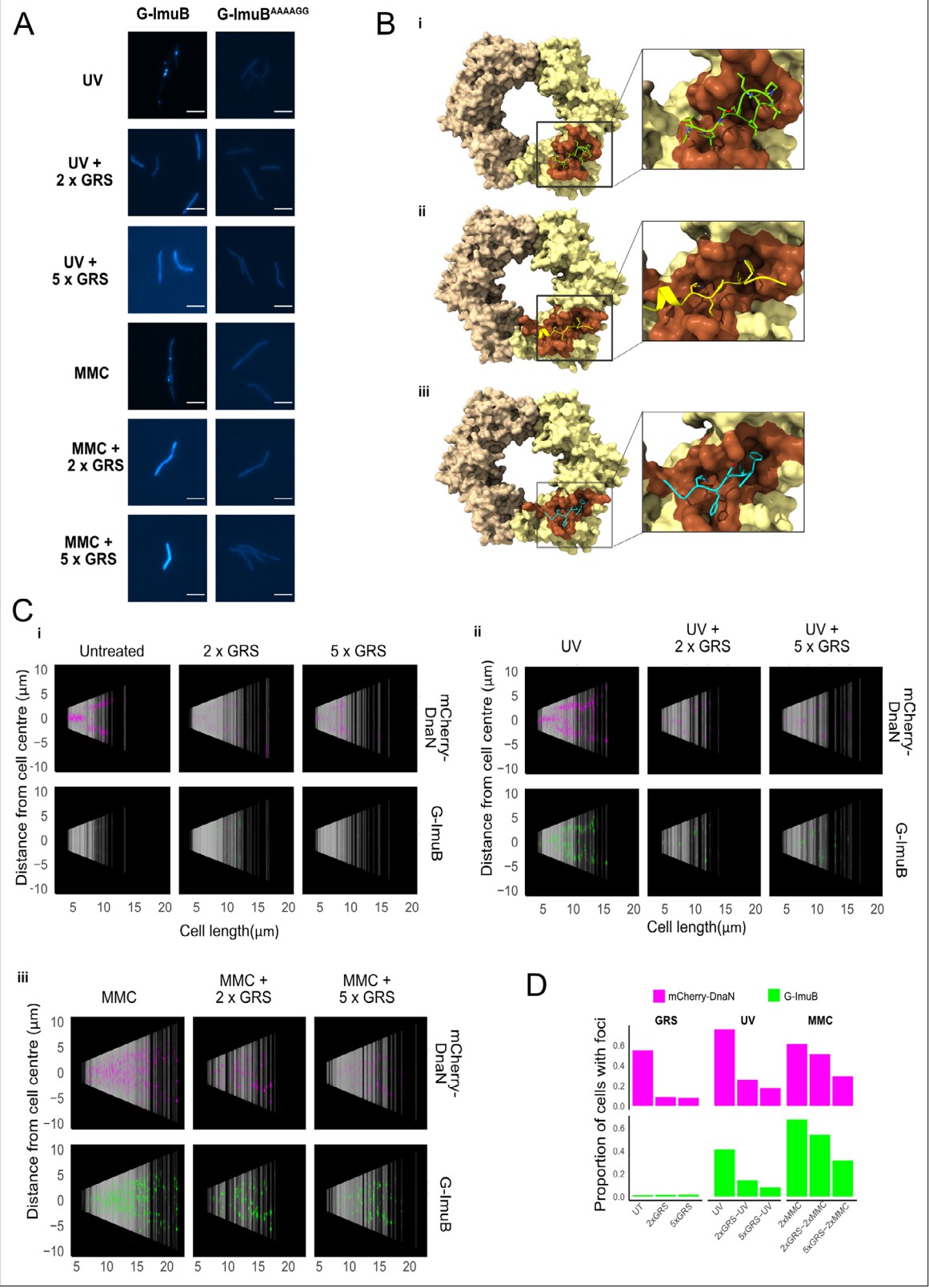

**Figure 5.** Disrupting the ImuB–β clamp interaction. (**A**) Representative images of G-ImuB exposed to 2× MIC mitomycin C (MMC) for 4 hr (top panel) or 2× MIC MMC plus griselimycin (GRS) for 4 hr (center panel), and the G-ImuB^AAAAGG mutant exposed to 2× MIC MMC for 4 hr (bottom panel). Scale bars, 5 µm. (**B**) Interactions of *M. smegmatis* β clamp with (**i**) GRS, (ii) ImuB, or (iii) the replicative DNA PolIIIα subunit, DnaE1. The interaction of the β clamp with GRS is represented by the X-ray structure of the complex (PDB id: 5AGU). Predicted interactions with ImuB (S347-I359) and DnaE1 (M947-G954)

*Figure 5 continued on next page*

*Figure 5 continued*

are derived from the respective AlphaFold models. Interacting peptides are shown as smoothed traces with side chains. The brown β clamp region indicates residues in contact with corresponding peptides. Molecular contacts were derived from 3D structures using the VoroContacts web server. Detailed contact data are provided separately (*Supplementary file 1* – DnaN Contact Data). (**C**) Cells aligned by mid-cell position, arranged according to cell length and colored (magenta – mCherry-DnaN foci, green – G-ImuB foci) according to fluorescence intensity, showing the presence of G-ImuB foci following MMC treatment and the lack of foci after GRS exposure. G-ImuB$^{AAAAGG}$ shows no foci after MMC treatment, similar to the G-ImuB strain following GRS exposure. (**D**) Proportions of cells containing either mCherry-DnaN or G-ImuB foci following exposure to GRS alone at 2× MIC or 5× MIC, or in combination with either single-dose ultra-violet (UV) or MMC at 2× MIC or 5× MIC. Additional Source data are available in Figure5.zip, accessible at http://doi.org/10.5061/dryad.76hdr7szc.

The online version of this article includes the following figure supplement(s) for figure 5:

**Figure supplement 1.** Comparison of *M. smegmatis* and *M. tuberculosis* β sliding clamps bound to griselimycin (GRS) crystal structures.

## Discussion

In *E. coli*, the DNA damage-induced SOS response triggers overexpression of *umuC*, *umuD*, and *recA* (*Maslowska et al., 2019*). UmuC is an error prone Y-family DNA polymerase that requires the binding of UmuD'$_2$, RecA, and ATP to reach full activity; this multi-protein 'mutasome', collectively referred to as DNA PolV, has been implicated in DNA damage tolerance and induced mutagenesis (*Goodman et al., 2016*). At the time of initiating the work reported here, genetic evidence from diverse bacteria lacking PolV homologs supported the co-dependent operation of ImuA, ImuB, and DnaE2 in the LexA-regulated SOS response, suggesting these proteins might function in an analogous manner (*McHenry, 2011*; *Ippoliti et al., 2012*). In mycobacteria, in which they have been individually implicated in DNA damage tolerance and induced mutagenesis (*Boshoff and Mizrahi, 2000*; *Warner et al., 2010*), the ImuA homolog, ImuA', replaces ImuA. Nevertheless, the inferred universal model for mutasome function in bacteria lacking an *E. coli* PolV homolog was the same (*Timinskas and Venclovas, 2019*): the catalytically inactive Y family polymerase, ImuB, functions as hub protein, interacting physically with the β clamp via a defined β clamp-binding motif and with DnaE2 and ImuA' (or ImuA) via unknown mechanisms which might include the ImuB C-terminal region or subregions thereof, including the RecA-NT motif (*Timinskas and Venclovas, 2019*). However, the absence of any direct biochemical and/or structural evidence to support the proposed protein interactions meant this assumption was speculative. Moreover, whereas *E. coli* PolV is known to be subject to multiple forms of regulation – including temporal (*Robinson et al., 2015*), spatial (*Robinson et al., 2015*), internal (*Erdem et al., 2014*), and conformational (*Jiang et al., 2009*; *Gruber et al., 2015*; *Jaszczur et al., 2019*) – the expression dynamics and subcellular localizations of the mycobacterial mutasome proteins were mostly unknown. Certainly, genomic organization alone (*imuA'–imuB/dnaE2* constitute a 'split' mutagenic cassette; *Erill et al., 2006*) could not predict the stoichiometry of any inferred protein complexes, nor the subcellular location(s) of individual mutasome components and their interacting partners.

By fluorescently tagging the known mutasome proteins, we have observed in real time the consistent formation of co-occurring ImuB–β clamp foci in mycobacterial cell populations exposed to genotoxic stress. Although less pronounced than ImuB, we also detected the frequent, reproducible co-occurrence of DnaE2 with the β clamp under the same conditions. Notably, recruitment of ImuB into foci occurred in mutants lacking functional DnaE2 or ImuA' but was prevented when the ImuB–β clamp-binding motif was mutated – apparently identifying the primacy of the ImuB–β clamp interaction in mutasome organization. In contrast, the function(s) and subcellular dynamics of ImuA' remain enigmatic: VFP-ImuA' consistently produced diffuse fluorescence in DNA-damaged bacilli, precluding any definitive insights into its potential association with ImuB (or DnaE2) in vivo. ImuA' and ImuB are encoded in a two-gene operon; therefore, their differential intracellular profiles (ImuB concentrated in foci, ImuA' diffusely distributed) were not predicted (nor predictable) based on genomic organization but instead required direct observation of the tagged proteins. Here, we note that these results have been reproduced by others in very recent work published during revision of our manuscript (*Ng et al., 2023*). The distinct intracellular distributions in vivo contrasted, too, with the biochemical analyses, in which the demonstrated co-elution of ImuA'–ImuB and ImuA'–ImuB–β clamp complexes provided important confirmation of the ImuA'–ImuB interaction inferred previously (*Warner et al., 2010*). Therefore, while difficult to reconcile with the in vitro data, the absence here of a clear co-localization signal in live cells might indicate the transient association of ImuA' with its mutasome partners

or, possibly, that a posttranslational modification is required in live bacteria – by analogy with the proteolytic cleavage of UmuD to UmuD' in the *E. coli* SOS response (*Goodman et al., 2016*). Future work will require single-molecule tracking of ImuA' to resolve this possibility.

The original identification of the *imuA–imuB–dnaE2* cassette noted its close association with LexA across diverse bacteria; that is, genomes containing the cassette invariably encoded a LexA homolog, too (*Erill et al., 2006*). Recent work in mycobacteria has added unexpected nuance to that regulatory framework, namely that the split *imuA'–imuB/dnaE2* cassette is subject to transcriptional control by both the 'classic' LexA/RecA-regulated SOS response and the PafBC-mediated DNA damage response (*Adefisayo et al., 2021*). The authors of that work also report that, while the two regulatory mechanisms are partially redundant for genotoxic stresses including UV and MMC exposure, fluoroquinolones appear to be specific inducers of PafBC only. In addition to suggesting that chromosomal mutagenesis is co-dependent on PafBC and SOS, these observations are important in identifying an apparent 'fail-safe' mechanism in mycobacteria in which the mutasome components are induced irrespective of DNA damage type – again reinforcing the centrality of these proteins in damage tolerance and, by implication, adaptive mutagenesis. Here, it is important to consider also the potential role of the mycobacterial DinB-type DNA polymerases in genome diversification in *M. tuberculosis* (*Dupuy et al., 2022*; *Dupuy et al., 2023*). Although expression of these Y family polymerases is not induced in *M. tuberculosis* in response to DNA damage (the *M. smegmatis* SOS response includes a third DinB homolog, DinB3, but this gene is absent from the *M. tuberculosis* genome), there is evidence suggesting some functional redundancy with DnaE2. Moreover, the differential capacity of *M. tuberculosis* DinB1 and DinB2 to bind the β clamp suggests the potential for complex protein interplay at stalled replication forks, the exact details of which remain to be elucidated.

We previously observed that the *imuB*$^{AAAAGG}$ β clamp-binding motif mutation eliminated UV-induced mutagenesis and MMC damage tolerance in *M. smegmatis* (*Warner et al., 2010*), phenocopying deletion of any of the three mutasome components (*imuA'*, *imuB*, and *dnaE2*) alone or in combination. Given the abrogation of ImuB focus formation, it seems reasonable to infer a direct link between ImuB–β clamp focus formation and mutasome function. In turn, this suggests that blockade of ImuB focus formation might offer a tractable read-out for a screen designed to identify mutasome inhibitors – a possibility reinforced by the observed co-elution in biochemical assays of β with ImuB and, separately, of the β clamp with pre-formed ImuA'–ImuB complexes. In this context, it was notable in this study that GRS disrupted the ImuB–β clamp interaction in vitro and prevented ImuB focus formation in mycobacteria treated simultaneously with MMC and GRS.

The discrepant complementation phenotypes observed for V-ImuA' and G-ImuB in the DNA damage tolerance (involving growth on MMC-containing solid media) versus induced mutagenesis (transient MMC or UV exposure during liquid culture) assays suggests that addition of the bulky fluorophore might have prevented full function of these mutasome proteins. Whereas UV irradiation predominantly generates cyclobutane dimers and pyrimidine–pyrimidone (6–4) photoproducts (*Franklin et al., 1985*), MMC induces a variety of different DNA lesions, including inter- and intrastrand cross-links. These are likely to require multiple repair pathways and, potentially, the interaction of mutasome components with additional protein partners – which might be prevented by the bulky fluorescent tags. The DnaE2–EGFP fusion proved the exception; in this context, it might be instructive to consider recent evidence implicating DnaE2 in gap filling following nucleotide excision repair in non-replicating *Caulobacter crescentus* cells (*Joseph et al., 2021*). These observations suggest the importance of identifying other potential interacting partners of mycobacterial DnaE2 (and the other mutasome components), work which is currently underway in our laboratory.

The potential for inhibitors of DNA replication to accelerate the development of genetic resistance through the induction of mutagenic repair/tolerance pathways (*Cirz et al., 2005*; *Barrett et al., 2019*; *Revitt-Mills and Robinson, 2020*) is a valid and commonly cited concern that might partially explain the relative under-exploration of DNA metabolism as source of new antibacterial drug targets (*Reiche et al., 2017*; *van Eijk et al., 2017*). Our results suggest that GRS could offer an interesting exception: that is, in binding the β clamp at the site of interaction with the DnaE1 replicative DNA polymerase as well as other DNA metabolizing proteins (*Kling et al., 2015*), including the clamp loader complex and ImuB, GRS appears to possess an intrinsic protective mechanism against induced mutagenesis – blocking both ImuB-dependent mutasome recruitment to stalled replisomes and post-repair fixation of mutations by the replicative polymerase, DnaE1. This 'resistance-proofing' capacity, which is

supported by the observed restriction of GRS resistance to low-frequency, high-fitness cost amplifications of the *dnaN* genomic region with very few to no 'off-target' single nucleotide polymorphisms (SNPs), might also contribute to the observed bactericidal effect of GRS against mycobacteria (*Kling et al., 2015*). In addition, it reinforces the β clamp as a vulnerable target for new TB drug development (*Bosch et al., 2021*). In this context, it is worth noting that inhibition of DnaE1 replicative polymerase function might represent a general solution to the problem of drug-induced (auto)mutagenesis by preventing fixation of repair/tolerance-generated mutations; in support of this inference, another natural product, nargenicin, which inhibits *M. tuberculosis* DnaE1 via a DNA-dependent mechanism, fails to yield spontaneous resistance mutations in vitro (*Chengalroyen et al., 2021*). Therefore, while the essentiality of DNA replication proteins such as DnaN and DnaE1 for mycobacterial viability poses a challenge to the design of assays to detect 'anti-evolution' compounds targeting these proteins (because inhibition of their essential, replicative function is growth inhibitory), GRS (and nargenicin) appear to provide compelling evidence that inhibition of some DNA replicative and repair functions might ameliorate the perceived risks in targeting this area of mycobacterial metabolism.

DnaE2-dependent DNA damage tolerance and induced mutagenesis were originally discovered using *M. smegmatis* as model mycobacterial organism, with key additional observations – including the contribution to pathogenicity and evolution of resistance under drug therapy – made in the pathogen, *M. tuberculosis* (*Boshoff et al., 2003*). Continuing that trend, the subsequent elucidation of the roles of ImuA' and ImuB as essential 'accessory factors' confirmed that the fundamentals of mutasome function were equivalent in both species (*Warner et al., 2010*). Our reliance in the current work on *M. smegmatis* as proxy is therefore justifiable, but does require caution in extrapolating the refined model for mutasome function to all other mycobacteria encoding mutasome proteins, including *M. tuberculosis*. That said, it is tempting to consider the implications of the results described here to an obligate pathogen whose persistence within its human host depends on the ability to drive successive cycles of infection, disease – in some cases latency followed by reactivation disease – and transmission (*Lin and Flynn, 2018*). Such cycles are inevitably vulnerable to multiple potential evolutionary culs-de-sac which might arise in consequence of the elimination of the bacillus by the host (clearance) or the demise of the organism within the infected individual (controlled subclinical infection, or host death). Modern *M. tuberculosis* strains therefore represent the genotypes that have successfully adapted to human colonization (*Gagneux, 2018*), evolving with their obligate host through changes in lifestyle and nutritional habits (with their associated implications for non-communicable diseases such as diabetes), the near-universal administration of the BCG vaccination, the emergence of the HIV co-pandemic, and the widespread use of frontline combination chemotherapy (*Warner et al., 2015*). While the emergence and propagation of drug-resistant isolates characterized by a variety of polymorphisms at multiple genomic loci (*Warner et al., 2017*; *Farhat et al., 2019*; *Payne et al., 2019*) provides strongest proof of the capacity for genetic variation in *M. tuberculosis*, other lines of evidence include the highly subdivided population structure of the *M. tuberculosis* complex (*Riojas et al., 2018*), the well-described geographical host–pathogen sympatry (*Hershberg et al., 2008*; *Brynildsrud et al., 2018*) and, more recently, the observation of intra-patient bacillary microdiversity (*Ley et al., 2019*). In combination, these elements support the ongoing evolution of *M. tuberculosis*, as well as suggest the potential that 'anti-evolution' therapeutics might yield much greater benefit in the clinical context than can be inferred from in vitro studies – in which the pressures on an obligate pathogen can only be approximated. That is, in addition to identifying the mutasome as target for adjunctive therapeutics designed to protect anti-TB drugs against emergent resistance, the results presented here support the further exploration of this and related strategies to disarm host-adaptive mechanisms in a major human pathogen and growing contributor to antimicrobial resistance.

## Materials and methods
### Bacterial strains and culture conditions

All mycobacterial strains (*Supplementary file 2* – Key Reagents) were grown in liquid culture containing Difco Middlebrook 7H9 Broth (BD Biosciences, San Jose, CA) and supplemented with 0.2% (vol/vol) glycerol (Sigma-Aldrich, St. Louis, MO), 0.005% (vol/vol) Tween 80 (Sigma-Aldrich, St. Louis, MO), and 10% (vol/vol) BBL Middlebrook OADC Enrichment (BD Biosciences, San Jose, CA). For *M. smegmatis*, liquid cultures were incubated at 37°C with orbital shaking at 100 rpm, until the desired growth

density was attained – measured by spectrophotometry at a wavelength of 600 nm – before further experimentation. Solid media comprised Difco Middlebrook 7H10 Agar (BD Biosciences, San Jose, CA) supplemented with 0.5% (vol/vol) glycerol (Sigma-Aldrich, St. Louis, MO), and 10% (vol/vol) BBL Middlebrook OADC Enrichment (BD Biosciences, San Jose, CA). Solid media plates were incubated at 37°C for 3–4 days or until colonies had formed.

## Mutasome reporter constructs

The V-*imuA*′ construct was designed by altering the coding sequence of *imuA*′ within the complementing vector, pAINT::*imuA*′ (*Warner et al., 2010*), so that the coding sequence of VFP (*Nagai et al., 2002*) was inserted in-frame after the start codon of the *imuA*′ ORF. Furthermore, an in-frame FLAG tag sequence (*Einhauer and Jungbauer, 2001*) was inserted between the coding region of *vfp* and *imuA*′ to produce a single ORF encoding VFP-FLAG-ImuA′. For ImuB, the construct PSOS(*imuA*′)-*egfp-imuB* was designed such that the regulatory elements immediately upstream of *imuA*′ were inserted immediately upstream of the *imuB* ORF which was further altered by inserting the sequence encoding EGFP (*Cormack et al., 1996*) linked to a FLAG tag-encoded sequence immediately after the start codon of *imuB* to produce a single ORF encoding EGFP-FLAG-ImuB′ which was cloned into pMCAINT::*imuB* (*Warner et al., 2010*). For DnaE2, the *egfp* sequence was inserted in-frame after the start codon of *M. smegmatis dnaE2* (*Figure 1—figure supplement 1A*).

## Mutant binding *G-imuB*^AAAAGG^ construct

To introduce the [352]AAAAGG[357] *imuB* allele (*Warner et al., 2010*) into the EGFP-ImuB protein, the nucleotide sequence from pMCAINT::*imuB*^AAAAGG^ was swapped into the corresponding position of PSOS(*imuA*′)-*egfp-imuB* to yield pMCAINT::PSOS(*imuA*′)-*egfp-imuB*^AAAAGG^.

## *M. smegmatis* mutasome reporter strains

*M. smegmatis* strain V-ImuA′ was generated by introducing the pAINT::*vfp-imuA*′ plasmid into Δ*imuA*′ (*Warner et al., 2010*) by the standard electroporation method. Strains G-ImuB, and G-ImuB^AAAAGG^ were developed by integration of the pMCAINT::PSOS(*imuA*′)-*egfp-imuB*, or pMCAINT::PSOS(*imuA*′)-*egfp-imuB*^AAAAGG^ plasmid, respectively, into the genome of Δ*imuB* (*Warner et al., 2010*). To generate the G-DnaE2 strain, pTweety::*egfp-dnaE2* was electroporated into Δ*dnaE2* (*Warner et al., 2010*). The *dnaN-mCherry*::G-*imuB*, *dnaN-mCherry*::V-*imuA*′, and *dnaN-mCherry*::G-*dnaE2* strains were developed by the electroporation of pMCAINT::PSOS(*imuA*′)-*egfp-imuB*, pAINT::*vfp-imuA*′ and pTweety::*egfp-dnaE2* into the *M. smegmatis dnaN-mCherry* background (*Santi et al., 2013*). Mutasome-deficient strains Δ*imuA*′, Δ*dnaE2*, and *dnaE2*^AIA^ were electroporated with pMCAINT::PSOS(*imuA*′)-*egfp-imuB* to produce Δ*imuA*′::*G-imuB*, Δ*dnaE2*::*G-imuB*, and *dnaE2*^AIA^::*G-imuB*, respectively.

## Antibiotic treatments

MMC (from *Streptomyces caespitosus*) (Sigma-Aldrich, St. Louis, MO) was dissolved in ddH$_2$O, while GRS was dissolved in dimethyl sulfoxide (DMSO). Cultures of *M. smegmatis* were grown in 7H9-OADC – supplemented with selection antibiotic where applicable – at 37°C to an optical density (OD$_{600}$) ~0.2–0.4. Thereafter, cultures were split into separate 5 ml cultures and MMC and/or GRS added to a final concentration dependent on the MIC (*Kling et al., 2015*).

## DNA damage sensitivity and mutagenesis assays

UV-induced mutagenesis assays were performed as previously described (*Boshoff et al., 2003*; *Warner et al., 2010*), with RIF-resistant colonies enumerated on solid media after 5 days of growth. For UV-induced DNA damage survival assays, *M. smegmatis* strains were grown in liquid culture to OD$_{600}$ ~ 0.5, following which a 10-fold dilution series was spotted (5 µl/spot) on standard 7H10 medium, allowed to dry, and then exposed to UV at 12.5 or 25 mJ/cm$^2$; plates were imaged after 3 days' incubation at 37°C. MMC-induced mutagenesis assays were performed by treating log-phase bacteria with 0.5× MIC MMC for 6 hr, following which the bacteria were washed and RIF-resistant colonies were enumerated on solid media as before. Mutation frequencies were calculated by dividing the number of RIF-resistant colonies of each sample by the CFU/ml of untreated sample. For MMC damage sensitivity assays, the cultures were grown to OD$_{600}$ ~ 0.4, following which a 10-fold dilution

series was spotted on standard 7H10 medium and 7H10 medium supplemented with MMC; plates were incubated for 3 days and imaged.

## Snapshot microscopy

Single snapshot micrographs of *M. smegmatis* cells were captured with a Zeiss Axioskop M, Zeiss Axio.Scope, and Zeiss Axio.Observer Z1. Briefly, 2.0–5.0 µl of liquid culture was placed between a No. 1.5 glass coverslip and microscope slide. A transmitted mercury lamp light was used together with filter cubes to visualize fluorescence using a ×100 1.4 NA plan apochromatic oil immersion objective lens. Samples were located using either transmitted light, differential interference contrast, or epifluorescence. Snapshot images were captured with either a Zeiss 1 MP or Zeiss AxioCam HRm monochrome camera. Images of the same experiment were captured with the same instrument and exposure settings. Green fluorescence of EGFP was detected using the Zeiss Filter Set 38 HE. Red fluorescence of mCherry was detected using the Zeiss Filter Set 43. Images were captured using Axio-Vision 4.7 or ZEN Blue Microscope and Imaging Software. Images were processed using Fiji (*Schindelin et al., 2012*); images of the same strain were contrasted to the same maximum and minimum within an experiment.

## Quantitative image analysis

*M. smegmatis* bacilli were plotted from shortest to longest and aligned according to their midcell position (0 on the *y*-axis) using the MicrobeJ plugin of ImageJ (*Ducret et al., 2016*). Along each point of the cell, a dot was generated and colored according to the fluorescence intensity along the medial axis of the bacillus. Therefore, this plot represents the fluorescence intensity along the medial axis of every bacillus imaged under the relevant experimental conditions. R was used for visual representation of the data.

## Single-cell time-lapse fluorescence microscopy

Liquid cultures of *M. smegmatis* reporter strains were grown to mid-logarithmic phase ($OD_{600}$ = 0.6), cells were collected by centrifugation at 3900 × *g* for 5 min and concentrated 10-fold in 7H9 medium. The cells were filtered through a polyvinylidene difluoride syringe filter (Millipore) with a 5-µm pore size to yield a clump-free cell suspension. The single-cell suspension was spread on a semi-permeable membrane and secured between a glass coverslip and the *serpentine 2 chip* (*Delincé et al., 2016*) in a custom-made PMMA/Aluminium holder (*Dhar and Manina, 2015*). Time-lapse microscopy employing a DeltaVision personalDV inverted fluorescence microscope (Applied Precision, WA) with a ×100 oil immersion objective was used to image single cells of *M. smegmatis*. The bacteria and microfluidic chip were maintained at 37°C in an environmental chamber with a continuous flow of 7H9 medium, with or without 100 ng/ml of MMC, at a constant flow rate of 25 µl/min, as described previously (*Wakamoto et al., 2013*; *Dhar and Manina, 2015*). Images were obtained every 10 min on phase-contrast and fluorescence channels (for EGFP, excitation filter 470/40 nm, emission filter 525/50 nm; for mCherry, excitation filter 572/35, emission filter 632/60; for YFP excitation filter 500/20 nm, emission filter 535/30 nm) using a CoolSnap HQ2 camera. Image-based autofocus was performed on each point prior to image acquisition. Experiments were repeated two to four times; a typical experiment collected images from up to 80 XY points at the 10-min intervals. The images were analyzed using Fiji (*Schindelin et al., 2012*).

## Protein expression and purification

N-terminally His-tagged *M. smegmatis* ImuB was co-expressed with ImuA' in *E. coli* BL21(DE3) cells using two expression vectors from the NKI-LIC vector suite (*Luna-Vargas et al., 2011*): pETNKI-his-3C-LIC-kan for ImuB and pCDFNKI-StrepII3C-LIC-strep for ImuA' that have different resistance markers, kanamycin and streptomycin; as well as different origins of replication, ColE1 and CloDF13, respectively. Protein production was induced with isopropyl 1-thio-β-D-galactopyranoside at 30°C for 2 hr. The ImuBA' complex was purified using a Histrap column followed by a Superdex 200 16/60 column. Both N-His6 *M. smegmatis* ImuB and β clamp were expressed in *E. coli* BL21(DE3) cells and purified using HisTrap, HiTrap Q, and S200 columns. All proteins were flash frozen in liquid nitrogen and stored at −80°C.

## Size-exclusion chromatography analysis

Samples of individual proteins and the different complexes were injected onto a PC3.2/30 (2.4 ml) Superdex 200 Increase gel filtration column (GE Healthcare) pre-equilibrated in 50 mM Tris pH 8.5 and 300 mM NaCl. Thereafter, 50 µl fractions were collected and analyzed by SDS–PAGE electrophoresis using 4–12% NuPage Bis-Tris precast gels (Life Technologies). Gels were stained with 0.01% (vol/vol) 2,2,2-trichloroethanol and imaged with UV light.

## Thermal unfolding experiments

Melting curves of the *M. smegmatis* β clamp (5 µM) in the presence and absence of GRS (15 µM) were measured in UV capillaries using the Tycho NT6 (NanoTemper Technologies) where the protein unfolding is followed by detecting the fluorescence of intrinsic tryptophan and tyrosine residues at both emission wavelengths of 350 and 330 nm.

## Structure modeling and analysis

Structural models for *M. segmatis* β clamp interaction with ImuB and DnaE1 were generated using ColabFold implementation (*Mirdita et al., 2022*) of AlphaFold-Multimer v.2 (*Jumper et al., 2021*; https://doi.org/10.1101/2021.10.04.463034). Structures of *M. smegmatis* and *M. tuberculosis* clamp complexes with GRS were obtained from PDB (PDB ids 5AH2 and 5AGU, respectively). Residues at the interaction interfaces were identified using VoroContacts server (*Olechnovič and Venclovas, 2021*). Structures were visualized using UCSF ChimeraX (*Goddard et al., 2018*).

## Acknowledgements

This work was supported by the US National Institute of Child Health and Human Development (NICHD) U01HD085531 (to DFW and RW). We acknowledge the funding support of the Research Council of Norway (R&D Project 261669 'Reversing antimicrobial resistance') (to DFW), the South African Medical Research Council (to VM and DFW); the National Research Foundation of South Africa (to DFW and VM); a Senior International Research Scholars grant from the Howard Hughes Medical Institute (to VM); and a LUMC Fellowship (to MHL). In addition, MAR is grateful to the South African National Research Foundation (NRF) for financial assistance during his PhD training (grant no. 104683) as well as the Whitehead Scientific Travel Award. ZAM is grateful to the University of Cape Town, the David and Elaine Potter Foundation Research Fellowship, and the Swiss Government Excellence Research Scholarship for financial assistance during her PhD.

## Additional information

### Funding

| Funder | Grant reference number | Author |
| --- | --- | --- |
| Eunice Kennedy Shriver National Institute of Child Health and Human Development | U01HD085531 | Roger Woodgate |
| Norges Forskningsråd | 261669 | Digby F Warner |
| South African Medical Research Council | SHIP and Extramural Unit | Valerie Mizrahi |
| National Research Foundation | | Valerie Mizrahi |
| Howard Hughes Medical Institute | Senior International Research Scholars | Valerie Mizrahi |
| Leids Universitair Medisch Centrum | LUMC Fellowship | Meindert H Lamers |
| National Research Foundation | 104683 | Michael A Reiche |

| Funder | Grant reference number | Author |
|---|---|---|
| David and Elaine Potter Foundation | PhD Fellowship | Zela Alexandria-Mae Martin |

The funders had no role in study design, data collection, and interpretation, or the decision to submit the work for publication.

## Author contributions
Sophia Gessner, Michael A Reiche, Conceptualization, Formal analysis, Investigation, Methodology, Writing – original draft, Writing – review and editing; Zela Alexandria-Mae Martin, Conceptualization, Investigation, Methodology, Writing – original draft, Writing – review and editing; Joana A Santos, Conceptualization, Investigation, Methodology, Writing – original draft; Ryan Dinkele, Formal analysis, Visualization; Atondaho Ramudzuli, Saber Anoosheh, Resources; Neeraj Dhar, Resources, Supervision, Investigation; Timothy J de Wet, Visualization; Dirk M Lang, John D McKinney, Resources, Supervision; Jesse Aaron, Teng-Leong Chew, Resources, Methodology; Jennifer Herrmann, Rolf Müller, Resources, Writing – review and editing; Roger Woodgate, Funding acquisition, Writing – review and editing; Valerie Mizrahi, Supervision, Writing – review and editing; Česlovas Venclovas, Formal analysis, Investigation, Methodology, Writing – review and editing; Meindert H Lamers, Conceptualization, Resources, Supervision; Digby F Warner, Conceptualization, Supervision, Funding acquisition, Methodology, Writing – review and editing

## Author ORCIDs
Sophia Gessner ⬚ http://orcid.org/0000-0003-0824-0079
Neeraj Dhar ⬚ https://orcid.org/0000-0002-5887-8137
Timothy J de Wet ⬚ http://orcid.org/0000-0002-3978-5322
Rolf Müller ⬚ http://orcid.org/0000-0002-1042-5665
John D McKinney ⬚ http://orcid.org/0000-0002-0557-3479
Roger Woodgate ⬚ https://orcid.org/0000-0001-5581-4616
Valerie Mizrahi ⬚ http://orcid.org/0000-0003-4824-9115
Meindert H Lamers ⬚ http://orcid.org/0000-0002-4205-1338
Digby F Warner ⬚ https://orcid.org/0000-0002-4146-0930

## Decision letter and Author response
Decision letter https://doi.org/10.7554/eLife.75628.sa1
Author response https://doi.org/10.7554/eLife.75628.sa2

# Additional files

## Supplementary files
• Supplementary file 1. DnaN contact data.
• Supplementary file 2. Key reagents.
• Transparent reporting form

## Data availability
Source data for all figures contained in the manuscript and SI have been deposited in Dryad (http://doi.org/10.5061/dryad.76hdr7szc).

The following dataset was generated:

| Author(s) | Year | Dataset title | Dataset URL | Database and Identifier |
|---|---|---|---|---|
| Warner DF | 2023 | Data from: The mycobacterial ImuA'-ImuB-DnaE2 mutasome: composition and recruitment in live cells | http://doi.org/10.5061/dryad.76hdr7szc | Dryad Digital Repository, 10.5061/dryad.76hdr7szc |

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
