## [Editor Report]

This important study investigates the localization dynamics of the mycobacterial mutasome complex, comprised of ImuA', ImuB, and DnaE2. The mutasome complex has a key role in promoting mutagenic DNA replication during stress to increase the mutation rate and potential for selection of drug resistant mutations. The authors provide compelling evidence that ImuB localizes with the β-clamp upon damage exposure and that the clamp binding motif in ImuB is essential for its localization. These studies lay the ground for future work in this area and will be intriguing to a broad audience interested in bacterial physiology.

---

## [Decision Letter]

**Decision letter after peer review:**

Thank you for submitting your article "The mycobacterial mutasome: composition and recruitment in live cells" for consideration by *eLife*. Your article has been reviewed by 3 peer reviewers, one of whom is a member of our Board of Reviewing Editors, and the evaluation has been overseen by Gisela Storz as the Senior Editor. The reviewers have opted to remain anonymous.

Essential revisions:

In this manuscript, the authors test their previously proposed model (also presented in Figure 1A) that ImuB interacts with the DnaN DNA polymerase III β clamp to recruit DnaE2. The previously identified mutasome components ImuA', ImuB, and DnaE2 and essential for DNA-damage induced mutagenesis. Although the exact function of ImuA' and ImuB is unknown, ImuB has long before been proposed to interact with DnaN via an interaction domain within ImuB that has already been identified. Since the experiments herein test and validate a well-establish model, the results are somewhat expected. However, all models should be tested experimentally, making this an important confirmation. The manuscript nicely makes use of both in vivo and in vitro approaches and the data is convincing for the most part, although the inability of the fusion proteins used in this study to complement the knockout strains during exposure to the DNA damaging agent MMC does raise an important limitation of the tools used herein and brings into question whether MMC should have been the genotoxic agent used in the studies. In addition, a major concern is the limited new biological insight gained from the study in its current form and a revision should address these limitations. The reviewers make several suggestions to help improve the scope and impact of the manuscript, as detailed here.

1) The section starting at Line 89 describes experiments expressing the fluorescently tagged proteins in M. smegmatis, but does not mention the mCherry-DnaN strand background even though the data shown in Figure 1 is in that background. Then a couple sections later starting on Line 174 the authors mention using the mCherry-DnaN background in a "new" set of experiments, but refer to the same data in Figure 1. This current organization is deceptive in that it makes it sound like these experiments were done twice, once in a background with mCherry-DnaN and once without. Were the experiments done in the absence of mCherry-DnaN? Is this what is in Figure 1 —figure supplement 1? If so, it would be better to include this in Figure 1 and would be beneficial to show that the localization of these proteins is not affected by the mCherry-DnaN tag. As written these sections basically just repeat each other almost word for word in terms of observations and should be combined up front. Regardless, imaging data without mCherry DnaN would be valuable.

2) The studies examining the effect of the fluorescently tagged proteins on mutagenesis and tolerance to DNA damage should be included in the main figure, particularly given the importance of the tagged constructs not complementing the deletions for survival following DNA damage. This is an important finding and highlights that localization of ImuB to foci is not sufficient to promote viability and function. It also looks like the DnaE2 construct did complement during MMC treatment, but the others did not. The authors should also include experiments with untagged versions to determine if it is the tag that is precluding complementation, rather than an expression issue or something with the vector.

3) Since the deletion if ImuA' affects MMC survival, could the fact that its localization remains disperse explain why these strains are not complementing? Since the DnaE2 construct does appear to complement, it could be interesting to test ImuA' localization in this strain.

4) Figure 1B is not referenced in the text. What time point is this? This information should also be included in the Figure Legend.

5) Does the δ imuA' affect DnaE2 localization?

6) The way the manuscript is written, it jumps around a lot, going to figure 4, then back to figure 3. Are there replicate experiments to present with appropriate controls so that this does not need to happen?

7) The ImuA' and ImuB translational fusions are not active in MMC-induced mutagenesis. Given that these translational fusions failed to support function, the contribution of the observed foci following MMC exposure to ImuABC function is unclear. Do these foci represent functional complexes, dead intermediates due to the fluorescent tags, or are they artefacts of inactive proteins? Given that MMC is the primary damage used by the authors, it is necessary to establish that the tags do not perturb lesion bypass (and mutagenesis) under this damage. The presented evidence would suggest that the tags do perturb mutasome function in MMC and that this treatment may not be reliable for extracting information on dynamics. In addition, the translational ImuA' and ImuB fusions supported UV-induced mutagenesis. Thus, UV seems to be a superior DNA damaging agent for the colocalization studies. Results using UV should be included to provide a more accurate picture of Imu-clamp colocalization.

8) 3D microscopy is not well explained for the non-expert. How was colocalization established along the z-axis? A better description for non-experts is needed. In addition, more quantification of the images is necessary. For example, It is also unclear what percentage cells have ImuB and DnaE2 foci, if these percentages vary in a dose-dependent manner, and if they reduce during recovery. In addition, how many cells (%) show co-localization of the different Imu proteins with β clamp? Do they always colocalize, and how often is an Imu protein or a clamp protein focus seen alone? If most foci show colocalization, do the authors conclude that all clamps on DNA are in complex with ImuB and would this be surprising? β-clamp foci denote all loaded replisomes, including the ones that are actively synthesizing DNA with the replicative polymerase. Thus, do the number of localizations observed for clamp vary with and without damage?

Might foci be comprised of multiple proteins (in addition to the different Imus and the clamp), some of which may be active on the DNA and others of which may be in close proximity and ready to be called into action? ImuB also interacts with Pol III (Warner et al., 2010, PNAS). Is Pol III also present in the complexes? Im Figure 1 supp 1B, do the number of localizations observed vary between UV and MMC treatment?

9) Does addition of GRS impact ImuA'BC-dependent mutagenesis? This would be the direct way to test the proposal that therapeutic disruption of ImuB-clamp interactions would inhibit mutagenesis in *M. tuberculosis*. Mutagenesis assays should be performed in the GRS-MMC treated conditions to support any conclusion on GRS abrogation of mutasome action.

10) The authors were unable to purify ImuC, so its interactions with ImuA' and ImuB, and the effects of these interactions on its polymerase activity are unknown. While this is unfortunate, it is not the fault of the authors. However, in the absence of ImuC, it seems there is more that could be done with ImuA' and ImuB to test and extend further the published model for ImuABC function (Warner et al., 2010, PNAS). For example, does ImuB interact with itself via its C-terminal domain as it did in yeast-two-hybrid? Does ImuA' interact with the β clamp, or influence the affinity of ImuB for the clamp? Does ImuB lack an intrinsic DNA polymerase activity as predicted by its lack of conserved acid active site residues (Warner et al., 2010, PNAS)? Related to this, are both ImuA' and ImuC dispensable for ImuB-clamp colocalization in live cells?

11) The authors seem to make two contradicting arguments – on the one hand, they argue that ImuA'BC acts on multiple DNA damaging agents, but on the other, they argue that different DNA damaging agents may require different accessory proteins for proper Imu function. The authors could reconcile these arguments early in the text.

12) What was the spontaneous *M. tuberculosis* mutation frequency, and how does it compare to the frequencies of UV-induced mutagenesis for the ∆imuA', ∆imuB, and ∆imuC strains?

13) In general, the temporal dynamics of mutasome association with clamp are a promising part of the manuscript, but the authors do not explore this thoroughly. Quantitative analysis, dose dependency and temporal dynamics during damage and recovery (as interpreted by the authors throughout the manuscript) are lacking characterization.

14) In Line 206- The authors state that their observations suggest that deficiency in ImuA' would affect ImuB localization, but it is not clear why they believe this to be true. ImuB has a clamp binding motif, so it is likely to associate with the clamp irrespective of ImuA'. In order to support ImuA'-related conclusion, the authors would need to quantify the number of localizations observed for ImuB and β clamp in the presence and absence of ImuA'.

15) GRS impact on clamp/ImuB: In Line 295 and Figure 3B. The authors should show the clamp+GRS alone profile as well.

16) Figure 4 and conclusions with regards to impact of GRS as well as ImuB clamp binding mutant. It is possible GRS alone affects clamp stability, irrespective of mutasome function. The authors need to image the clamp after GRS treatment, to assess whether the lack of ImuB localization is because it cannot bind the clamp or because the clamp itself is no longer localized.

17) What is the impact of the clamp-binding-ImuB mutant on clamp localization? Is it similar to GRS treatment?

Other comments:

1. L139 – L143: To make this conclusion, authors need to carry out experiments across a range of doses.

2. Figure 2 – individual fluorescence panels need to be shown independently. Currently it is hard to visualize the localizations with accuracy.

3. L276 in Mtb, the PafBC regulatory system also influences Imu expression. The authors need to rephrase.

4. Please use β-clamp everywhere (and not only B).

5. Figure 3A. The difference in elution profiles between ImuA'-ImuB-clamp an ImuB-clamp would suggest that ImuB-clamp interaction alone might be less stable (in absence of ImuA'). Could the authors comment on the same?

6. Figure 4 legend. Top and bottom panel references need to be updated.

7. L314-315: Details of population analysis are missing in the legends or text.

8. L341. "when" instead of "where".

9. L338. "repair/ tolerance" pathways.

10. L379. Could the authors clarify? Do they envision DnaE2 acting without clamp? In that case, what could the potential mechanism be?

11. L405. It is unclear whether the GRS phenotype is due to its action on the replisome, independent of damage / mutasome effects, unless the impact of GRS on clamp alone is tested.

12. It is understandable that the authors use M. smeg as a model system to derive conclusions on action of the mutasome. However, the paragraph starting L410 needs to be toned down as all experiments are performed in smeg and not M.tb. Any extrapolation of their conclusions need to be explicitly stated to the reader.

---

## [Author Response]

Essential revisions:In this manuscript, the authors test their previously proposed model (also presented in Figure 1A) that ImuB interacts with the DnaN DNA polymerase III β clamp to recruit DnaE2. The previously identified mutasome components ImuA', ImuB, and DnaE2 and essential for DNA-damage induced mutagenesis. Although the exact function of ImuA' and ImuB is unknown, ImuB has long before been proposed to interact with DnaN via an interaction domain within ImuB that has already been identified. Since the experiments herein test and validate a well-establish model, the results are somewhat expected. However, all models should be tested experimentally, making this an important confirmation. The manuscript nicely makes use of both in vivo and in vitro approaches and the data is convincing for the most part, although the inability of the fusion proteins used in this study to complement the knockout strains during exposure to the DNA damaging agent MMC does raise an important limitation of the tools used herein and brings into question whether MMC should have been the genotoxic agent used in the studies.

The reviewers justifiably questioned the appropriateness of mitomycin C (MMC) as genotoxic agent given the observed inability of the fluorescent reporter alleles to complement the respective mutasome deletion mutants under extended MMC exposure in the DNA damage survival assay. To address this concern, we have generated new data showing that V-ImuAʹ and G-ImuB fully complement the corresponding deletion mutants in the MMC damage-induced mutagenesis assay (Figure 2). We also show that UV treatment recapitulates the mutagenesis results observed with MMC; that is, the fluorescent reporter alleles complement loss of the wildtype mutasome proteins under UV-induced mutagenesis (Figure 1 —figure supplement 1). In combination, these data appear sufficient to support the notion that the fluorescent reporters are functional in DNA damageinduced mutagenesis.

Our original observation that V-ImuAʹ and G-ImuB fail to complement the corresponding deletion mutants in the MMC DNA damage survival assay holds. Although the precise reason for this difference remains elusive, we think it might be instructive in revealing that the addition of fluorescent tags to these mutasome proteins has possibly disrupted an interaction(s) with another protein(s) that, under prolonged exposure to MMC *–* a clastogen which causes multiple types of DNA lesion, is critical for DNA damage survival. Work is ongoing to resolve this conundrum.

1) The section starting at Line 89 describes experiments expressing the fluorescently tagged proteins in M. smegmatis, but does not mention the mCherry-DnaN strand background even though the data shown in Figure 1 is in that background. Then a couple sections later starting on Line 174 the authors mention using the mCherry-DnaN background in a "new" set of experiments, but refer to the same data in Figure 1. This current organization is deceptive in that it makes it sound like these experiments were done twice, once in a background with mCherry-DnaN and once without. Were the experiments done in the absence of mCherry-DnaN? Is this what is in Figure 1 —figure supplement 1? If so, it would be better to include this in Figure 1 and would be beneficial to show that the localization of these proteins is not affected by the mCherry-DnaN tag. As written these sections basically just repeat each other almost word for word in terms of observations and should be combined up front. Regardless, imaging data without mCherry DnaN would be valuable.

We apologize for the confusing presentation of Results in the original version. The fluorescent reporters were introduced and visualized in both the respective knock-out backgrounds as well as in the mCherry-DnaN background. As requested, Figure 1 has been revised to contain images obtained from the fluorescent reporters in the knock-out backgrounds alone; this is clarified in the revised figure legend. All subsequent experiments were performed in the mCherry-DnaN background; this is made explicit from the section entitled "ImuB localizes with the *dnaN*-encoded β clamp following DNA damage" and thereafter.

2) The studies examining the effect of the fluorescently tagged proteins on mutagenesis and tolerance to DNA damage should be included in the main figure, particularly given the importance of the tagged constructs not complementing the deletions for survival following DNA damage. This is an important finding and highlights that localization of ImuB to foci is not sufficient to promote viability and function.

We agree with this suggestion and, as noted in response to reviewer comment 1, we have revised the manuscript to incorporate new data, as well as restructuring the Results section to foreground these observations. The results of the complementation assays are now included in a new Figure 2, which contains original data from the UV-induced mutagenesis (Figure 2A) and MMC damage sensitivity (Figure 2C) assays, and new data from a MMC damage-induced mutagenesis assay (Figure 2B).

It also looks like the DnaE2 construct did complement during MMC treatment, but the others did not. The authors should also include experiments with untagged versions to determine if it is the tag that is precluding complementation, rather than an expression issue or something with the vector.

In this work, we utilized the same complementation system as reported previously in Warner *et al.* (2010; doi:10.1073/pnas.1002614107), which restored full function using untagged ImuAʹ, ImuB, and DnaE2 alleles (see Figure 1B in that paper); all we did in the current study was append the fluorescent tags to the respective mutasome genes. It seems very unlikely, therefore, that issues with the expression system might account for the lack of complementation in the MMC damage sensitivity assays. Consistent with related comments (see response to introductory reviewers’ comment), our interpretation instead is that the presence of the fluorescent tags might have impacted full protein function in cells under prolonged exposure to a lethal genotoxic stress. This conclusion is supported by the functionality of the tagged alleles in both UV- and MMCinduced mutagenesis assays (Figure 2).

3) Since the deletion if ImuA' affects MMC survival, could the fact that its localization remains disperse explain why these strains are not complementing? Since the DnaE2 construct does appear to complement, it could be interesting to test ImuA' localization in this strain.

This is an excellent question. We cannot exclude the possibility that it is the presence of the VFP tag that prevents ImuA’ association with ImuB in vivo in live mycobacterial cells, resulting in the diffuse fluorescence signal. However, the biochemical data presented in Figure 4 —figure supplement 1B indicate that the addition of fluorescent tags to either ImuB or ImuA’ do not disrupt the interaction of these proteins in vitro. Moreover, (see response to introductory reviewers’ comment) the V-ImuA’ allele does complement DNA damage-induced mutagenesis following exposure to either MMC or UV, suggesting functionality of the tagged protein. Work is underway in our laboratory to elucidate the components of the mutasome in live cells without requiring fluorescent labels.

4) Figure 1B is not referenced in the text. What time point is this? This information should also be included in the Figure Legend.

Figure 1B has been removed in preparing the revised manuscript.

5) Does the δ imuA' affect DnaE2 localization?

This is another very good question. As shown in Figure 1, unlike ImuB, which produces distinct foci, the DnaE2 signal is less definitive. Therefore, addressing the potential roles of the other mutasome proteins in DnaE2 localization requires alternative, more sensitive approaches such as single-molecule tracking. This work is ongoing and, as noted below, includes attempts to express purified DnaE2 for analysis in biochemical assays alone and in combination with the other mutasome components.

6) The way the manuscript is written, it jumps around a lot, going to figure 4, then back to figure 3. Are there replicate experiments to present with appropriate controls so that this does not need to happen?

We apologize for the confusion caused by the previous presentation of the Results. As noted in response to reviewers’ comment 1, the revised version has been restructured in accordance with the reviewers’ recommendations to make it clearer and more logical. All results presented in the manuscript reflect multiple biological replicates utilizing appropriate controls.

7) The ImuA' and ImuB translational fusions are not active in MMC-induced mutagenesis. Given that these translational fusions failed to support function, the contribution of the observed foci following MMC exposure to ImuABC function is unclear. Do these foci represent functional complexes, dead intermediates due to the fluorescent tags, or are they artefacts of inactive proteins? Given that MMC is the primary damage used by the authors, it is necessary to establish that the tags do not perturb lesion bypass (and mutagenesis) under this damage. The presented evidence would suggest that the tags do perturb mutasome function in MMC and that this treatment may not be reliable for extracting information on dynamics. In addition, the translational ImuA' and ImuB fusions supported UV-induced mutagenesis. Thus, UV seems to be a superior DNA damaging agent for the colocalization studies. Results using UV should be included to provide a more accurate picture of Imu-clamp colocalization.

This comment reflects a common concern among all reviewers about the validity of making conclusions based on the MMC phenotypes alone. The revised version submitted here contains new data showing that V-ImuAʹ and G-ImuB fully complement the corresponding deletion mutants in the MMC damage-induced mutagenesis assay (Figure 2B) [Note that the first sentence in the reviewers’ comment, namely “The ImuA' and ImuB translational fusions are not active in MMC-induced mutagenesis”, is therefore not correct; the translational fusions are active in the MMC-induced mutagenesis assay, but not the MMC DNA damage survival assay.]

In addition, we present evidence that UV treatment recapitulates the results observed with MMC; that is, the fluorescent reporter alleles complement loss of the respective wildtype mutasome proteins in UV-induced mutagenesis (Figure 2A), confirming that this effect is not limited to a single DNA damaging agent. We also show that ImuB foci are induced following exposure of cells to both UV and MMC (for MMC, timelapse data confirm that this is dynamic and occurs rapidly following treatment; Figure 3 and Video 1). In previous work (Warner et al., 2010; doi:10.1073/pnas.1002614107), we reported that mutations in the ImuB β clamp-binding motif (QLPLWG) inhibited the ImuB-β clamp interaction in yeast two-hybrid assays (Figure 4 in Warner et al., 2010; doi:10.1073/pnas.1002614107), and abrogated UV-induced mutagenesis in live cells (Figure 6 in Warner et al., 2010; doi:10.1073/pnas.1002614107). Now, in this revised submission, we demonstrate that the imuB^AAAAGG^ allele disrupts ImuB-β clamp focus formation (Figure 5A), linking focus formation to induced mutagenesis. In combination, these observations suggest that the formation of ImuB foci is a common feature of the mycobacterial DNA damage response and is essential for mutasome function.

As indicated in earlier comments, the reason(s) for the discrepant results between the induced mutagenesis assays and survival assays remains unclear – and might reflect disruption of an interaction with an additional partner(s) required for surviving constant MMC exposure, or could be a consequence of the tag impairing optimal protein function under sustained, lethal genotoxic pressure during prolonged incubation on solid media. That said, the ability of the fusion proteins to support induced mutagenesis following MMC treatment suggests that the MMC-induced foci, like the UV-induced foci, are mutagenic.

As recommended by the reviewers, we have included UV treatment in the microscopy analyses; these results have added value to this work and are presented in Figure 3D. The nature of UV experiments unfortunately complicates the use of timelapse microscopy to monitor responses in UV-treated cells; for this reason, timelapse videos are only available for MMC-treated mycobacteria. Nevertheless, single timepoint fluorescent microscopy images are presented which indicate that UV exposure elicits an equivalent response to that observed following MMC treatment (Figure 1A).

8) 3D microscopy is not well explained for the non-expert. How was colocalization established along the z-axis? A better description for non-experts is needed. In addition, more quantification of the images is necessary. For example, It is also unclear what percentage cells have ImuB and DnaE2 foci, if these percentages vary in a dose-dependent manner, and if they reduce during recovery. In addition, how many cells (%) show co-localization of the different Imu proteins with β clamp? Do they always colocalize, and how often is an Imu protein or a clamp protein focus seen alone? If most foci show colocalization, do the authors conclude that all clamps on DNA are in complex with ImuB and would this be surprising? β-clamp foci denote all loaded replisomes, including the ones that are actively synthesizing DNA with the replicative polymerase. Thus, do the number of localizations observed for clamp vary with and without damage?

We thank the reviewer for this comment which prompted an overhaul of our presentation and analyses of the microscopy data. The descriptions of the microscopy and the associated analyses have been modified to make them more accessible to the non-expert. In addition, to address the valid request for quantitative analyses, several new figures have been included:

i) Figure 3D shows the position of mCherry-DnaN foci in relation to G-ImuB foci in cells expressing both fluorescent proteins; these results indicate strong ImuB-β clamp correlation in cells exposed to either UV or MMC, and near absent expression of ImuB in undamaged cells.

ii) Figure 3 —figure supplement 1 includes new data showing the proportion of cells with mCherry-DnaN and G-ImuB foci. As noted earlier, the less distinct signal obtained from GDnaE2 precluded similar analyses of potential DnaE2 correlations.

iii) Figure 5C and Figure 5D present data on the effect of griselimycin treatment on b clamp and ImuB focus formation in cells exposed to either MMC or UV, as well as in the absence of DNA damage.

Might foci be comprised of multiple proteins (in addition to the different Imus and the clamp), some of which may be active on the DNA and others of which may be in close proximity and ready to be called into action? ImuB also interacts with Pol III (Warner et al., 2010, PNAS). Is Pol III also present in the complexes? Im Figure 1 supp 1B, do the number of localizations observed vary between UV and MMC treatment?

The reviewers raise several very interesting questions here. The potential that other proteins, including the *dnaE1*-encoded DNA Polymerase IIIa subunit, might be present in foci has been subject of much discussion among the authors – especially given the importance of understanding how the switch occurs from the replicative polymerase to the translesion synthesis machinery. Studies to address these questions are ongoing in our laboratories, and no solid data are available yet to support informed speculation.

To address the query about whether the number of foci differs as a function of genotoxic stress (MMC or UV), we have included an additional analysis (Figure 3—figure supplement 1) showing the proportion of cells with foci as well as the proportion of cells that present only a single signal (that is, either mCherry-DnaN or G-ImuB) or both in response to the different DNA damaging treatments.

9) Does addition of GRS impact ImuA'BC-dependent mutagenesis? This would be the direct way to test the proposal that therapeutic disruption of ImuB-clamp interactions would inhibit mutagenesis in M. tuberculosis. Mutagenesis assays should be performed in the GRS-MMC treated conditions to support any conclusion on GRS abrogation of mutasome action.

The reviewers raise a valid point about the need for an experiment demonstrating that disruption of the ImuB-b clamp interaction inhibits mutagenesis. As noted in response to reviewers’ comment 7, this has been demonstrated genetically: the *imuB*^AAAAGG^ allele which disrupts ImuB-β clamp focus formation (Figure 5A) was shown previously to abrogate UV-induced mutagenesis in live cells (Figure 6 in Warner *et al.*, 2010; doi:10.1073/pnas.1002614107). The question, though, is whether this can be demonstrated pharmacologically – as the reviewers’ comment implies. Again, this is something we have discussed at length but have not been able to identify a suitable approach: although the reviewers suggest performing mutagenesis assays in the presence of griselimycin, this is not definitive since griselimycin targets both DNA replication – by preventing the DnaE1-b clamp interaction (Kling *et al.*, 2015; doi:10.1126/science.aaa4690) – and mutagenic DNA repair, by preventing ImuB-b clamp focus formation (this manuscript), as well as preventing b clamp localization through the likely inhibition of the clamp loader complex. Griselimycin-mediated replisome collapse has been described previously (Trojanowski *et al.*, 2019; *Antimicrob Agents Chemother*. doi:10.1128/AAC.00739-19).

Given that subsequent rounds of mycobacterial replication are required for fixation of induced mutations in the genome, it is not possible to separate the effects of replication inhibition from blocked mutagenesis – a conundrum which highlights the need for an on-target inhibitor of the mutasome (a DnaE2 inhibitor, for example). In the absence of a suitable test system, we present microbiological evidence (Figure 5) indicating that treatment of cells with griselimycin eliminates ImuB-β clamp focus formation under mutagenic DNA damage (MMC exposure) conditions, and back this up with biochemical data (Figure 4) showing disruption of ImuB-DnaN and ImuA’-ImuBDnaN interactions in assays utilizing the purified proteins. Together, these observations support the inference that inhibiting mutasome recruitment and/or function will substantively eliminate DNA damage-induced mutagenesis in mycobacteria.

10) The authors were unable to purify ImuC, so its interactions with ImuA' and ImuB, and the effects of these interactions on its polymerase activity are unknown. While this is unfortunate, it is not the fault of the authors. However, in the absence of ImuC, it seems there is more that could be done with ImuA' and ImuB to test and extend further the published model for ImuABC function (Warner et al., 2010, PNAS). For example, does ImuB interact with itself via its C-terminal domain as it did in yeast-two-hybrid? Does ImuA' interact with the β clamp, or influence the affinity of ImuB for the clamp? Does ImuB lack an intrinsic DNA polymerase activity as predicted by its lack of conserved acid active site residues (Warner et al., 2010, PNAS)? Related to this, are both ImuA' and ImuC dispensable for ImuB-clamp colocalization in live cells?

The reviewers have raised multiple very interesting questions and identified numerous potential lines of experimental inquiry. We agree that many of these are pressing and would enhance our understanding of the mutasome; in fact, we are actively pursuing these and many other questions currently. However, we respectfully submit that all are beyond the scope of the current study. The question about the potential roles of ImuA’ and DnaE2 (ImuC) in ImuB-b clamp localization is partially addressed in this manuscript: Figure 3 —figure supplement 2 shows that ImuB focus formation occurs despite the absence of ImuA’ or DnaE2, while Figure 4Aii presents biochemical data to support the ability of ImuB to interact directly with the b clamp in the absence of ImuA’ or DnaE2. Although these results do not directly interrogate the impact of either protein on ImuB-b clamp localization, the strong phenotype of ImuB-b clamp focus formation in DNA damaged cells suggests that the interaction is likely independent of other proteins.

11) The authors seem to make two contradicting arguments – on the one hand, they argue that ImuA'BC acts on multiple DNA damaging agents, but on the other, they argue that different DNA damaging agents may require different accessory proteins for proper Imu function. The authors could reconcile these arguments early in the text.

We apologize for this apparent contradiction which arose from clumsy sentence construction. To clarify, we have modified the text to read (L160-4):

"Although mutasome components are expressed in response to genotoxic stress arising from a variety of different sources, it is possible the different types and/or extent of DNA damage induced in the two separate assays used here (induced mutagenesis vs. DNA damage tolerance) might require distinct interactions with a different partner protein(s) and, further, that one/more of these might have been disrupted by the presence of the fluorescent tag(s)."

12) What was the spontaneous M. tuberculosis mutation frequency, and how does it compare to the frequencies of UV-induced mutagenesis for the ∆imuA', ∆imuB, and ∆imuC strains?

The *M. tuberculosis* mutation rate has previously been reported as 2-3 x 10^-10^ mutations per cell per generation (Boshoff *et al.*, 2003; doi:10.1016/s0092-8674(03)00270-8). The same study reported that the spontaneous mutation rate was unaffected in a *dnaE2* deletion mutant; however, UV-induced mutagenesis was effectively eliminated in the *dnaE2* knockout. In follow-up work (Warner *et al.*, 2010; doi:10.1073/pnas.1002614107), we demonstrated that deletion of any of the mutasome components (ImuA’, ImuB, or DnaE2), alone or in combination, eliminated UV-induced mutagenesis in both *M. smegmatis* and *M. tuberculosis*: that is, the UV-induced mutation frequencies in the mutasome mutants were indistinguishable from non-DNA-damaged strains.

13) In general, the temporal dynamics of mutasome association with clamp are a promising part of the manuscript, but the authors do not explore this thoroughly. Quantitative analysis, dose dependency and temporal dynamics during damage and recovery (as interpreted by the authors throughout the manuscript) are lacking characterization.

The reviewers have again identified a very interesting area of study which demands considerable additional experimental work. The heterogeneity evident in DNA damaged cells (see the different timelapse movies, for example) hints at the complexity inherent in tracking the recovery of individual cells. To tackle this problem, we are currently exploring an alternative microfluidic system that might enable the spatial constriction of mycobacterial cells necessary for extended single-cell tracking and analysis of sub-cellular protein localizations.

14) In Line 206- The authors state that their observations suggest that deficiency in ImuA' would affect ImuB localization, but it is not clear why they believe this to be true. ImuB has a clamp binding motif, so it is likely to associate with the clamp irrespective of ImuA'. In order to support ImuA'-related conclusion, the authors would need to quantify the number of localizations observed for ImuB and β clamp in the presence and absence of ImuA'.

The reviewers are correct in suggesting that the capacity of ImuB to bind the β clamp directly might be expected to obviate any impact of ImuA’ on ImuB localization. That said, the structural homology of ImuA' to RecA, plus the demonstrated essentiality of ImuA’ for mutasome function, meant we couldn’t dismiss a priori the possibility that ImuAʹ might affect ImuB localization, even indirectly. Two observations motivated this thinking: (i) knowledge that, in the *E. coli* system, RecA activates UmuDC in forming the functional DNA Polymerase V mutasome, and (ii) recent work indicating that ImuA inhibits the recombinase activity of RecA1 in *Myxococcus xanthus*, facilitating mutagenesis (Sheng *et al.*, 2021; *Appl Environ Microbiol*. doi:10.1128/AEM.00919-21).

Although we did not expect mycobacterial ImuA’ to fulfil analogous roles, we needed to exclude either possibility. If it were the case that ImuA’ (or DnaE2) influenced ImuB localization, we would expect to observe no or limited ImuB focus formation in the *imuAʹ* (or *dnaE2*) deletion mutant, which was not the case (Figure 3 —figure supplement 2). This result was also critical in establishing the centrality of the ImuB-β clamp interaction in mutasome function, in turn suggesting a potential high-throughput assay for novel chemical inhibitors of this essential protein-protein interaction.

15) GRS impact on clamp/ImuB: In Line 295 and Figure 3B. The authors should show the clamp+GRS alone profile as well.

Kling *et al.* (2015; doi:10.1126/science.aaa4690) demonstrated the binding of griselimycin to the b clamp in biochemical and structural assays; this observation is reiterated in the new modelling data (Figure 5B, Figure 5 —figure supplement 1) which indicate clearly that the region of griselimycin binding on the mycobacterial β clamp subunit overlaps with the region predicted to interact with other β clamp-binding proteins, including DnaE1 and ImuB.

16) Figure 4 and conclusions with regards to impact of GRS as well as ImuB clamp binding mutant. It is possible GRS alone affects clamp stability, irrespective of mutasome function. The authors need to image the clamp after GRS treatment, to assess whether the lack of ImuB localization is because it cannot bind the clamp or because the clamp itself is no longer localized.

The reviewers raise an important point which we realize was not adequately enunciated in our original submission. In their study, Kling *et al.* (2015; doi:10.1126/science.aaa4690) noted that griselimycin binds the β clamp at the site of multiple protein interactions, likely including the clamp-loader complex, too – an observation reinforced in our own modelling data (Figure 5B, Figure 5 —figure supplement 1) which extends the list of proteins utilizing this site to include ImuB. As the reviewers state, the effect would be to disrupt β clamp localization *independent* of any effect on ImuB. We agree with this interpretation, evidence for which is provided in Figure 5C where the disruptive impact of griselimycin treatment on mCherry-DnaN localization is clearly apparent. Moreover, griselimycin-mediated replisome collapse has been demonstrated previously (Trojanowski *et al.*, 2019; *Antimicrob Agents Chemother*. doi:10.1128/AAC.00739-19).

Our intention in the present work is not to claim that griselimycin specifically reduces mutasome function but to provide proof of concept evidence supporting the potential that an on-target inhibitor of the mutasome might offer a valuable addition to current antimycobacterial drugs by decreasing the capacity for DNA damage-induced mutagenesis in *M. tuberculosis*. We further argue (L439-49) that these observations might alleviate some of the concerns commonly associated with targeting (essential) DNA replication and repair proteins, namely the potential for accelerated/induced (auto)mutagenesis.

17) What is the impact of the clamp-binding-ImuB mutant on clamp localization? Is it similar to GRS treatment?

See response to reviewers’ comment 16, griselimycin binds the β clamp at the site of multiple protein interactions, including that which enables interaction of the β clamp with the clamp-loader complex. In contrast, the β clamp-binding-defective ImuB mutant (*imuB*^AAAAGG^) is not expected to affect β clamp loading. Therefore, this specific question was not experimentally addressed. It is worth noting, though, that analysis of the timelapse series featuring G-ImuB and mCherry-DnaN (Figure 3A, 3D) and the corresponding source data strongly suggests recruitment of ImuB to the β clamp foci; as noted earlier (see response to reviewer comment 13), elucidating the temporal dynamics of protein recruitment and localization is a key area of ongoing study.

Other comments:1. L139 – L143: To make this conclusion, authors need to carry out experiments across a range of doses.

This comment seems to question the inference that expression and recruitment of mutasome components occurs in response to a variety of DNA damaging conditions. As noted earlier (see response to reviewers’ comment 1), the revised manuscript includes experimental data utilizing the fluorescent reporter alleles in the respective knock-out backgrounds as well as in the mCherry-DnaN background, and following exposure to two concentrations of MMC (1×MIC, 0.5×MIC) and to UV irradiation. These observations, together with evidence that *imuA*’/*imuB* expression is co-regulated by both PafBC and SOS pathways in mycobacteria and induced following fluoroquinolone-mediated gyrase inhibition (Adefisayo *et al.*, 2021; *Nucleic Acids Res.* doi:10.1093/nar/gkab1169), suggest the conclusion is reasonable and supported by multiple independent lines of evidence.

2. Figure 2 – individual fluorescence panels need to be shown independently. Currently it is hard to visualize the localizations with accuracy.

As noted (see response to reviewers’ comment 8), we have restructured the Results to improve presentation of the microscopy data, including the addition of quantitative analyses. The original Figure 2 has been incorporated into a revised Figure 3 which includes quantitative analyses of protein localization data (Figure 3D). We note, too, that the timelapse movies are available as Video files, with all original data accessible via the open access Dryad data platform.

3. L276 in Mtb, the PafBC regulatory system also influences Imu expression. The authors need to rephrase.

This is a good point; we apologize for this inaccuracy which has been corrected in the revised version. The sentence (L294-6) now reads:

“Therefore, to test the prediction that the recruitment of EGFP-ImuB and mCherry-DnaN into discernible foci was dependent on the ImuB-β clamp protein-protein interaction, we introduced an *egfp*-*imuB*^AAAAGG^ allele (*G-imuB*^AAAAGG^) into the D*imuB* mutant.”

4. Please use β-clamp everywhere (and not only B).

Corrected throughout the revised manuscript.

5. Figure 3A. The difference in elution profiles between ImuA'-ImuB-clamp an ImuB-clamp would suggest that ImuB-clamp interaction alone might be less stable (in absence of ImuA'). Could the authors comment on the same?

Although tempting to speculate a role for ImuA’ in stabilizing the ImuB-β clamp, we have no additional evidence to support any definitive statements about this possibility.

6. Figure 4 legend. Top and bottom panel references need to be updated.

Figure 4 has been replaced with a new Figure 5.

7. L314-315: Details of population analysis are missing in the legends or text.

As noted, the original Figure 4 has been replaced with a new Figure 5, including expanded figure legend.

8. L341. "when" instead of "where".

Revised as requested.

9. L338. "repair/ tolerance" pathways.

Revised as requested.

10. L379. Could the authors clarify? Do they envision DnaE2 acting without clamp? In that case, what could the potential mechanism be?

This is an excellent question. The absence of a clear β clamp-binding motif in DnaE2, plus the inferred lack of a DnaE2-β clamp interaction in yeast two-hybrid assays (Warner *et al.*, 2010; doi:10.1073/pnas.1002614107), suggests that DnaE2 must operate without the β clamp. How this is accomplished is not clear, we can only assume that our understanding of DnaE2 function will be enhanced if/ when expression of the purified recombinant DnaE2 protein is achieved.

11. L405. It is unclear whether the GRS phenotype is due to its action on the replisome, independent of damage / mutasome effects, unless the impact of GRS on clamp alone is tested.

This comment echoes the concern raised previously (see reviewers’ comment 16) which arises from the effectively pleiotropic action griselimycin exerts on multiple replication and repair functions owing to the fact that it binds in the hydrophobic cleft between domains II and III of the β clamp (Kling *et al.*, 2015; doi:10.1126/science.aaa4690), the same interaction site utilized by multiple DNA metabolic proteins including ImuB (Figure 5B, Figure 5 —figure supplement 1). Indeed, our own data (see Figure 5C) demonstrating the disruptive impact of griselimycin treatment on mCherry-DnaN localization support the reviewers’ call for a definitive assay of chemically disrupted mutasome function. The problem, as we noted in our response to reviewer comment 9, is that griselimycin targets both DNA replication – by preventing the DnaE1-b clamp interaction (Kling *et al.*, 2015; doi:10.1126/science.aaa4690) – and mutagenic DNA repair, by preventing ImuB-b clamp focus formation (this manuscript), as well as preventing b clamp localization through the likely inhibition of the clamp loader complex. For this reason, it is not possible to separate the effects of replication inhibition from blocked mutagenesis – a conundrum which, we argued, reinforces the need for an on-target inhibitor of the mutasome. In the absence of such a compound, we have presented microbiological evidence (Figure 5) indicating that treatment of cells with griselimycin eliminates ImuB-β clamp focus formation under mutagenic DNA damage (MMC exposure) conditions, and we complemented this observation with biochemical data (Figure 4) showing disruption of ImuB-DnaN and ImuA’-ImuBDnaN interactions in assays utilizing the purified proteins. Based on these results, and previous work showing that genetic disruption of the ImuB-β clamp interaction eliminates UV-induced mutagenesis, we have argued that inhibiting mutasome recruitment and/or function will substantively eliminate DNA damage-induced mutagenesis in mycobacteria.

12. It is understandable that the authors use M. smeg as a model system to derive conclusions on action of the mutasome. However, the paragraph starting L410 needs to be toned down as all experiments are performed in smeg and not M.tb. Any extrapolation of their conclusions need to be explicitly stated to the reader.

The reviewers’ point is well made. We have revised the concluding paragraph (L45082) to reflect the reliance of the current study on *M. smegmatis* as mycobacterial model, and to encourage caution in extrapolating the results to *M. tuberculosis*.